# Towards Understanding Factual Knowledge of Large Language Models

**Xuming Hu[1,2*], Junzhe Chen[1*], Xiaochuan Li[1*], Yufei Guo[1], Lijie Wen[1†],**
**Philip S. Yu[3], Zhijiang Guo[4†]**
[1] Tsinghua University   [2] The Hong Kong University of Science and Technology (Guangzhou)
[3] University of Illinois at Chicago   [4] University of Cambridge
xuminghu@hkust-gz.edu.cn, wenlj@tsinghua.edu.cn, zg283@cam.ac.uk

## Abstract

Large language models (LLMs) have recently driven striking performance improvements across a range of natural language processing tasks. The factual knowledge acquired during pretraining and instruction tuning can be useful in various downstream tasks, such as question answering, and language generation. Unlike conventional Knowledge Bases (KBs) that explicitly store factual knowledge, LLMs implicitly store facts in their parameters. Content generated by the LLMs can often exhibit inaccuracies or deviations from the truth, due to facts that can be incorrectly induced or become obsolete over time. To this end, we aim to explore the extent and scope of factual knowledge within LLMs by designing the benchmark Pinocchio. Pinocchio contains 20K diverse factual questions that span different sources, timelines, domains, regions, and languages. Furthermore, we investigate whether LLMs can compose multiple facts, update factual knowledge temporally, reason over multiple pieces of facts, identify subtle factual differences, and resist adversarial examples. Extensive experiments on different sizes and types of LLMs show that existing LLMs still lack factual knowledge and suffer from various spurious correlations. We believe this is a critical bottleneck for realizing trustworthy artificial intelligence. The dataset Pinocchio and our codes are publicly available at: https://github.com/THU-BPM/Pinocchio.

## 1 Introduction

Large language models (LLMs) have revolutionized natural language processing (NLP) in recent years since they have significantly improved performance on various downstream tasks (Brown et al., 2020; Chowdhery et al., 2022; Ouyang et al., 2022; Touvron et al., 2023a;b; OpenAI, 2022; 2023). Prior efforts have shown that language models can store factual knowledge and act as knowledge bases (Petroni et al., 2019; Jiang et al., 2020c). Factual knowledge in language models acquired during pretraining can benefit knowledge-intensive downstream tasks such as question answering and fact checking (Roberts et al., 2020; Yu et al., 2023a; Pan et al., 2023).

Despite advancements in LLMs, they still struggle with generating content that exhibits inaccuracies or deviations from the facts and making reasoning errors (Lin et al., 2022; Bubeck et al., 2023). These factual errors can be difficult to identify since LLMs implicitly memorize facts through their parameters rather than explicitly store factual knowledge as traditional Knowledge Bases. Accessing and interpreting the computations and memories of these models can be challenging (Ribeiro et al., 2016; Belinkov & Glass, 2019), especially when APIs are the only means of interaction and many interpretation methods rely on weights and representations (Cao et al., 2021b). The presence of errors in stored factual knowledge or the incorrect induction and obsolescence of certain facts over time may be contributing factors to this limitation, which in turn affects the performance of LLMs (Elazar et al., 2021; Cao et al., 2021a). This limitation restricts the application of LLMs in some high-stakes areas, such as healthcare, finance, and law (Dong et al., 2022). Hence, exploring the degree to which LLMs hold factual information and their ability to reason with such knowledge is vital.

---

[*] Equal Contribution.
[†] Corresponding authors.

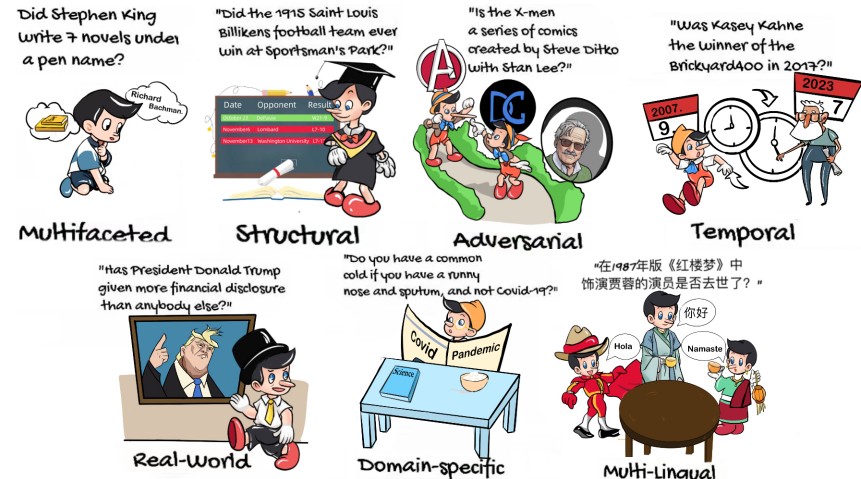

Figure 1: Pinocchio is a comprehensive dataset that tackles 7 distinct tasks related to factual knowledge and reasoning. It consists of 20,713 multiple-choice questions that have been sourced from various reliable and diverse channels.

To this end, we propose the Pinocchio, a testbed aimed at understanding factuality and reasoning for LLMs. It contains 20K diverse factual questions that span different sources, timelines, domains, regions, and languages. Furthermore, we investigate whether LLMs are able to recognize the combination of multiple facts, reason over structured and unstructured evidence, realize facts change over time, identify subtle factual differences, and resist adversarial examples based on the dataset. We control for problem difficulty in each distinct reasoning task to enable fine-grained analysis.

With the Pinocchio benchmark, we explore whether various LLMs (Scao et al., 2022b; Zhang et al., 2022; Ouyang et al., 2022; Chung et al., 2022; Touvron et al., 2023a; Chiang et al., 2023) could store factual knowledge and perform reasoning based on it. We envision Pinocchio as a suite of benchmarks, subsets of which could be separately utilized to assess certain model abilities of interest and analyze important strengths and limitations of LLMs. For instance, in temporal tasks, we find that LLMs lack factual knowledge for up-to-date questions; in complex factual tasks that require multi-hop reasoning, LLMs still have limitations, even when various prompting strategies are employed. We hope Pinocchio can serve as the initial step towards understanding the abilities of LLMs from multiple dimensions and facilitate the development of LLMs.

## 2 DATASET CONSTRUCTION

### 2.1 TASKS

Aiming to systematically evaluate the factual knowledge and related reasoning abilities of LLMs, we raise seven research questions, then carefully select factual statements from different sources summarized in Table 1.

- **Task 1: Multifaceted** Previous research (Petroni et al., 2019) has shown that small language models like BERT have the ability to retain relational knowledge from training data and answer "fill-in-the-blank" cloze statements. This raises the question of *whether LLMs can also store and reason over multiple pieces of facts obtained during pretraining*. It is not just important for LLMs to memorize individual facts accurately, but to also recognize and generate new combinations of facts from different sources. To investigate this issue, we have selected claims from the FEVER dataset (Thorne et al., 2018), which were written by human annotators based on information from Wikipedia articles. These claims are either supported or refuted by multiple facts from (the same or several) Wikipedia articles, or there is insufficient information available to verify them. To assess the performance of language models in handling various combinations of facts, we have sampled statements that require different numbers of evidence, ranging from one to many, enabling fine-grained analysis.
- **Task 2: Structural** In addition to unstructured text, factual knowledge is also commonly stored in a structured format, such as tables, lists, or databases (Bhagavatula et al., 2013). However,

Table 1: Pinocchio Dataset Sources, Descriptions, and Data Distribution.

| Domain | Description | Sources | Distribution | | | |
|--------|-------------|---------|------|-----------|-----|-----|
| | | | Fact. | Non-Fact. | NEI | ALL |
| Multifaceted | Contain multiple facts | FEVER | 1,111 | 1,111 | 1,110 | 3,332 |
| Structural | Contain structured and unstructured facts | FEVEROUS | 1,741 | 1,953 | 250 | 3,944 |
| Adversarial | Contain facts edited by adversarial methods | Symmetric, FM2 | 815 | 921 | - | 1,736 |
| Temporal | Contain facts that change over time | VitaminC | 1,898 | 1,043 | 355 | 3,296 |
| Real-World | Contain factual statements spread online | PolitiFact | 986 | 1,987 | 609 | 3,582 |
| Domain-Specific | Contain facts from health and science domains | PubHealth, SciFact | 1,156 | 715 | 737 | 2,608 |
| Multi-Lingual | Contain facts in different languages | XFact, CHEF | 820 | 848 | 547 | 2,215 |

current LLMs are primarily trained on unstructured text using next word prediction loss (Brown et al., 2020; Touvron et al., 2023a). In order to process structured data, it is often converted into text strings using various methods, such as linearizing tables. This raises the question of *whether LLMs are capable of effectively memorizing and reasoning over facts from structured sources, similar to their performance with unstructured text*. To investigate this question, we sample factual statements from the FEVEROUS dataset (Aly et al., 2021), which is constructed in a similar manner to FEVER but includes evidence in the form of tables, sentences, or both.

- **Task 3: Adversarial** Language models are known to be vulnerable to adversarial examples that are strategically modified to deceive even advanced models with hardly noticeable changes (Shen et al., 2023). Given this knowledge, it is important to examine *whether LLMs can withstand adversarial examples in the context of factuality*. To investigate this, we utilize two datasets, namely Symmetric (Schuster et al., 2019) and FM2 (Eisenschlos et al., 2021). These datasets consist of adversarial examples that have been crafted using various strategies, including temporal inference and diverting to unrelated facts.

- **Task 4: Temporal** Facts are not static but rather possess a dynamic nature. With the vast amount of new information constantly emerging, facts often undergo changes, additions, or alterations. It raises the question of *whether LLMs are able to adapt to these factual changes over time*. In particular, we wonder if LLMs are capable of discerning factual knowledge from different time periods, since the pretraining corpus may not be processed and organized chronologically. To explore this, we utilize the VitaminC (Schuster et al., 2021) dataset, which consists of claims based on modifications made to factual content in Wikipedia articles. Claims can be either refuted by outdated facts or supported by updated facts.

- **Task 5: Real-World** In contrast to other tasks that assume Wikipedia has all the essential factual information, verifying viral claims on the internet often requires not only factual knowledge from various sources but also common sense and worldly knowledge. An important query we have is *whether LLMs can effectively integrate diverse types and sources of knowledge acquired during training*. To address this, we select claims from the FactCheck (Misra, 2022) dataset, which consists of claims spread over the Internet and subsequently verified by journalists.

- **Task 6: Domain-Specific** In addition to the tasks mentioned earlier, which primarily focus on factual knowledge in general domains, we are also interested in exploring *how LLMs possess the capability to access domain-specific factual knowledge*. The domain-specific setting presents unique challenges. Take the science domain as an example, LLMs need to acquire background knowledge, handle quantitative reasoning, and comprehend specialized statistical language. To investigate this further, we sample claims from PubHealth (Kotonya & Toni, 2020) in the public health domain and SciFact (Wadden et al., 2022) in the science domain.

- **Task 7: Multi-Lingual** Existing LLMs are mainly trained on English corpus because of their abundance and quality (Chowdhery et al., 2022; Touvron et al., 2023a). However, the scarcity of training data in other languages raises the question of *whether LLMs can transfer the factual knowledge acquired in English to other languages*. To investigate this, we collected claims from various languages including French, Chinese, and more, using the XFACT dataset (Gupta & Srikumar, 2021) and the CHEF dataset (Hu et al., 2022b) in a total of 27 different languages.

## 2.2 ANNOTATION AND QUALITY CONTROL

Multiple-choice questions offer a practical approach to assess the complex capabilities of LLMs, of which GPT-4 is a prime example (OpenAI, 2023). Key benchmarks such as MMLU (Hendrycks et al., 2021b), HellaSwag (Zellers et al., 2019), ARC (Clark et al., 2018a), and TruthfulQA (Lin et al., 2022), all of which utilize multi-choice formats, serve distinct purposes in evaluating various aspects of GPT-4's proficiency. Specifically, the MMLU gauges an LLM's knowledge breadth and depth.

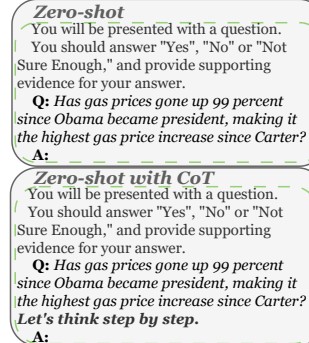

Figure 2: Illustration of prompts using different settings.

HellaSwag tests commonsense reasoning, and ARC focuses on challenging questions. TruthfulQA measures how LLMs mimic human falsehoods. Furthermore, the evaluation of language generation brings its own set of challenges, as a universal metric for measurement is currently lacking (Sai et al., 2023), which multiple-choice questions help to mitigate by offering straightforward classification accuracy for assessment (Hendrycks et al., 2021b). Also, prior studies (Kadavath et al., 2022) underscore that LLMs demonstrate reliable calibration on multiple-choice scenarios. Therefore, we also used the multi-choice questions as a simple but good proxy to evaluate the abilities of LLMs.

For data annotation, we hired 10 undergraduate students, all with good English proficiency. We asked the students to rewrite the original claims into questions without distorting factuality while providing factuality labels for the questions. By transforming declarative statements into questions, using a Question-Answering approach can more effectively elicit factual knowledge from LLMs (Kadavath et al., 2022; Lin et al., 2022), and we also illustrate through experiments in Sec. 4.2. Note that claims in the original datasets are usually labeled based on given evidence, e.g. evidence supports or refutes the claim, but in Pinocchio, we only need to judge the factuality of the question. So we use unified labels: Yes, No, Not Sure Enough. The three labels correspond respectively to Factual, Non-Factual, and Not Enough Information for factual questions. Considering that all fact-checking datasets use a three-label system (Guo et al., 2022), we did not modify the number of labels to maintain consistency in labeling. When dealing with factuality questions in low-resource languages, for Chinese, the 5 undergraduate students we hired are native Chinese speakers. For other low-resource languages, we first use Google Translate to translate them into English and generate factuality questions, then translate the English questions back to the corresponding languages. The label distribution is shown in Table 1. We paid the annotators accordingly based on the quantity and quality of the annotations.

We ensure the quality of the annotated factuality questions in two ways. The two authors of this paper served as meta-reviewers, sampling 10 questions from each of the three categories across the seven domains in Pinocchio. The meta-reviewers judged if the factuality labels were correct. For the 210 factuality questions, the average label accuracy was 92.4%. We divided the 10 students into two groups and had each group re-annotate a random 200 questions annotated by the other group, then calculated inter-annotator agreement (IAA). The final IAA was 85.6%. Based on meta-reviewer results and IAA, the factuality labels in Pinocchio are of good quality.

## 3 METHODOLOGY

### 3.1 MODELS

To give a comprehensive view of the status of LLMs in a factual context, we evaluate 10 accessible LLMs, undergone different training stages including pretraining, instruction tuning, and reinforcement learning from human feedback (Ouyang et al., 2022), covering diverse organizations and varying in size. A detailed description can be found in Appendix A.2.

### 3.2 PROMPT STRATEGY

As illustrated in Figure 2, we employ 4 types of prompts to elicit desired responses from LLMs, namely: Zero-shot, Zero-shot with CoT (Kojima et al., 2022), Few-shot, and Few-shot with CoT (Wei et al., 2022). Specifically, we begin by providing the model with task instruction, denoted as $Z$: "You

Table 2: Results obtained using different forms of prompts on 10 accessible LLMs.

| Methods | Zero-shot w/o CoT | | Zero-shot w/ CoT | | Few-shot w/o CoT | | Few-shot w/ CoT | | Overall Performance | |
|---|---|---|---|---|---|---|---|---|---|---|
| | Accuracy | F1 | Accuracy | F1 | Accuracy | F1 | Accuracy | F1 | Accuracy | F1 |
| OPT-6.7B | — | — | — | — | 36.9 | 27.9 | 37.9 | 28.5 | 18.8 | 14.3 |
| BLOOM-7B | 29.7 | 26.2 | 14.8 | 18.1 | 29.7 | 28.1 | 6.6 | 12.2 | 20.2 | 21.2 |
| LLaMA-7B | 31.8 | 29.6 | 22.3 | 24.9 | 36.8 | 28.6 | 35.3 | 31.4 | 31.6 | 28.6 |
| Alpaca-7B | 40.2 | 23.7 | 33.7 | 24.4 | 37.9 | 24.9 | 39.4 | 26.2 | 37.8 | 24.8 |
| Vicuna-7B | 33.2 | 33.6 | 34.2 | 32.9 | 35.5 | 34.8 | 48.5 | 40.6 | 37.9 | 34.9 |
| Vicuna-13B | 42.6 | 35.6 | 44.0 | 36.9 | 47.0 | 38.6 | 47.0 | 42.5 | 45.2 | 38.4 |
| ChatGLM-6B | 37.4 | 31.0 | 36.5 | 31.7 | 41.6 | 37.9 | 42.9 | 37.5 | 39.6 | 34.5 |
| Flan-T5-11B | 24.6 | 21.5 | 29.9 | 29.3 | 25.9 | 23.7 | 38.4 | 38.4 | 29.7 | 26.9 |
| Text-Davinci-002 | 45.2 | 36.2 | 45.7 | 37.3 | 46.6 | 40.4 | 46.2 | 42.5 | 45.9 | 39.1 |
| Text-Davinci-003 | 42.8 | 41.4 | 43.1 | 42.1 | **48.8** | 43.2 | 46.9 | 43.4 | 45.5 | 42.5 |
| GPT-3.5-Turbo | **46.9** | **44.3** | 46.8 | **44.4** | 47.2 | **44.7** | 47.1 | 45.7 | **47.0** | **44.8** |

will be given a question. You should answer whether it is Yes, No, or Not Sure Enough and show your evidence". This instruction informs the LLMs about the expected input and output. Subsequently, for any given input $Q$, we anticipate obtaining an output label $Y$ from the LLMs $f$: $Y = f(Q, Z)$.

**Zero-Shot Prompt** In the zero-shot setting, the LLMs are expected to provide answers based on the Question $Q$ and the task instruction $Z$. We anticipate that the LLMs can directly generate the factual answer "No" when presented with $Q$: "Has gas prices gone up 99 percent since Obama became president, making it the highest gas price increase since Carter?" The zero-shot with CoT setting extends the question $Q$ by adding a two-stage prompt (Kojima et al., 2022): "Let's think step by step", designed to encourage the LLMs to contemplate the process of determining the factual label $Y$.

**Few-Shot Prompt** In the few-shot setting, we employ three shots for model input ($Q$). Detailed examples of the prompts in Figure 2 are presented in Appendix A.4. In the few-shot with CoT setting, we provide potential reasoning instructions to the LLMs before presenting the factual label ($Y$). As shown in Figure 2, for the $Q$: "Is there a capital called Mogadish?" Our reasoning approach entails first explaining the noun phrase in the $Q$ (the subject and object), and subsequently elaborating on modifying phrases such as predicates or adjectives. Regarding the subject "Mogadish", we begin by furnishing a detailed definition: "Mogadishu is a city in East Africa, specifically in Somalia." Following this, we proceed to reason about the relation between "Mogadish" and "capital": "Furthermore, the capital of Somalia is indeed Mogadishu." Consequently, we arrive at the ultimate factual label: "Therefore, the answer is Yes."

## 4 EXPERIMENTS

In an effort to take the initial step in understanding the capabilities of LLMs, we undertake a comprehensive analysis of various LLMs on Pinocchio, under different conditions and tasks.

### 4.1 MAIN RESULTS

In Table 2, we present the average results of 10 accessible LLMs operating under varying settings on Pinocchio, run three times each. From Table 2, we draw the following conclusions:

- Regarding overall performance, we observe that, on average, LLMs without instruction tuning underperform those with instruction tuning by 16.0%. GPT family LLMs undergoing RLHF exhibit superior results, indicating that instruction tuning and RLHF optimize alignment with human knowledge, thereby improving factual question response accuracy.
- Results obtained using the Few-shot setting significantly outperform those obtained when simply asking factual questions to LLMs in the Zero-shot setting, especially for models without RLHF, exhibiting an average improvement of 7.3%. This highlights the capability of some sample prompts to better extract the inherent factual knowledge of LLMs.
- Using the CoT method, we observed a relative boost in performance in LLMs subjected to instruction tuning and RLHF, improving by an average of 2.1%. Notably, the factual accuracy of LLMs like OPT, BLOOM, and LLaMA was mostly stable or even decreased. A review of outputs from these untuned LLMs revealed that, post-CoT application, LLMs tend to produce related

Table 3: Results of different LLMs using Few-shot w/ CoT prompts across different tasks.

| Task | Multifaceted | | Structural | | Adversarial | | Temporal | | Real-World | | Domain Specific | | Multi-lingual | |
|---|---|---|---|---|---|---|---|---|---|---|---|---|---|---|
| | Acc. | F1 | Acc. | F1 | Acc. | F1 | Acc. | F1 | Acc. | F1 | Acc. | F1 | Acc. | F1 |
| OPT-6.7B | 34.5 | 24.1 | 45.5 | 30.9 | 51.8 | 51.7 | 30.0 | 18.0 | **53.7** | 27.5 | 28.2 | 28.3 | 16.2 | 17.7 |
| BLOOM-7B | 10.7 | 13.5 | 0.8 | 3.5 | 2.0 | 3.7 | 3.7 | 7.7 | 5.4 | 8.5 | 11.8 | 15.6 | 9.8 | 15.9 |
| LLaMA-7B | 38.3 | 33.9 | 44.1 | 32.1 | 43.2 | 46.1 | 41.6 | 30.0 | 26.4 | 26.3 | 23.6 | 25.0 | 27.8 | 27.7 |
| Alpaca-7B | 38.6 | 28.8 | 48.0 | 23.6 | 46.4 | 35.1 | 49.6 | 26.1 | 24.5 | 19.9 | 42.9 | 26.8 | 24.2 | 17.7 |
| Vicuna-7B | 44.2 | 36.0 | 49.7 | 36.3 | 59.0 | 59.2 | **50.1** | 37.6 | 49.0 | 41.8 | 44.3 | 38.6 | **46.7** | 43.1 |
| Vicuna-13B | 49.9 | 45.3 | 48.1 | 37.9 | 58.9 | 60.0 | 45.4 | 37.8 | 47.7 | 42.7 | 43.5 | 40.4 | 37.8 | 37.9 |
| ChatGLM-6B | 41.0 | 36.0 | 46.8 | 35.7 | 51.5 | 48.6 | 39.4 | 32.4 | 48.9 | 34.8 | 35.2 | 35.0 | 37.1 | 35.3 |
| Flan-T5-11B | 49.2 | 49.4 | 43.5 | 33.7 | 54.7 | 56.6 | 31.6 | 30.6 | 31.1 | 29.4 | 35.6 | 34.6 | 25.3 | 14.4 |
| Text-Davinci-002 | 47.7 | 47.7 | **50.8** | 38.4 | 64.2 | 64.3 | 33.9 | 31.1 | 51.7 | 41.4 | 36.4 | 36.1 | 43.1 | 39.5 |
| Text-Davinci-003 | 51.1 | 47.8 | 44.3 | 33.7 | 64.1 | 63.7 | 41.4 | 35.1 | 48.0 | 42.8 | 40.4 | 41.4 | 43.7 | **43.6** |
| GPT-3.5-Turbo | 53.6 | **53.1** | 44.8 | 37.8 | 67.4 | 67.4 | 37.4 | 33.9 | 50.4 | 43.1 | 38.7 | 40.3 | 41.3 | 41.1 |

content considerations, and extensive considerations often overshadow factual discernment tasks, causing incorrect factual label outputs. In contrast, for instruction-tuned LLMs, the CoT method facilitates enhanced exploration of factual entity relations in questions, resulting in accurate factual labels. See Appendix A.5 for detailed case analyses.

- The OPT model, without being tuned to instructions, struggles significantly to output correct factual labels under the settings of Zero-shot and Zero-shot CoT, often resulting in either a repetition of the original question or a refusal to output any content at all. This issue is somewhat alleviated under the settings of Few-shot and Few-shot CoT.

- Additionally, we studied the hyperparameters of LLMs. Due to limited computing resources, we only explored Vicuna-7B and Vicuna-13B. We found that as model parameters increase, performance on factual questions improves correspondingly, with an average increase of 5.4%. This indicates that LLMs with more parameters can store more world knowledge and have stronger factual knowledge recognition capabilities.

In Table 3, we present the factual performance of LLMs in various tasks under the Few-shot CoT setting. This reveals the relative difficulty LLMs have in understanding and responding to factual questions in different tasks, providing insights for future training of factual knowledge in LLMs. From Table 3, it is observed that LLMs exhibit relatively poorer performance on factual questions related to the real-world, domain-specific knowledge, and multilingualism, being on average 6.4% lower compared to the other four tasks. This is attributed to the fact that the training data for LLMs typically come from general domains and are not up-to-date, which indirectly inspires the exploration of retrieval-augmented LLMs (Ram et al., 2023). We analyze the LLMs in different tasks in Sec. 4.2.

## 4.2 ANALYSIS

In this section, we explore LLMs' capabilities focusing on key areas like handling of multi-hop factual questions, proficiency in diverse prompt strategies, and tackling challenges like numerical reasoning and entity ambiguity. We also examine their performance on time-sensitive factual questions, against adversarial attacks, with fine-grained labels and prompts in multiple languages.

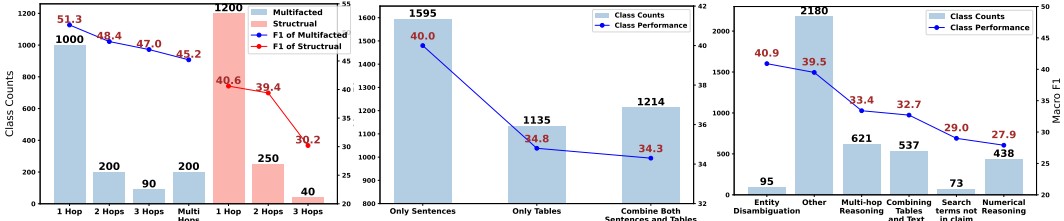

(a) Multi-hop Reasoning Analysis  (b) Structural Knowledge Analysis (c) Challenges of Different Questions

Figure 3: GPT-3.5-Turbo's outcomes across three distinct tasks under Few-shot CoT setting.

**Multi-hop Factual Question Analysis** To analyze the performance of LLMs when faced with factual questions based on multiple pieces of facts that require complex logical reasoning, we categorize multifaced and structural factual questions into distinct subsets, depending on the number of "hops" necessary to validate each factual question. To maintain fairness, we randomly sampled 1,490 data pieces from each of the two datasets for verification. Figure 3(a) illustrates the data

counts and Macro F1 scores of GPT-3.5-Turbo for each respective subset. The figure reveals a clear pattern: as the number of "hops" increases, the reasoning chain for deriving conclusions from existing factual knowledge extends, necessitating heightened logical reasoning capabilities from the LLMs. Consequently, the performance of the LLMs exhibits diminishing trends.

**Structural Knowledge Analysis in LLMs**    To investigate whether LLMs can effectively memorize factual knowledge from structured data, we divided the structural task questions into three subsets according to evidence distribution: evidence in unstructured data (Only text), structured data (Only tables), or both (Combine text and tables). Figure 3(b) shows a notable decline (Avg. -5.5%) in GPT-3.5-Turbo's performance when evidence involves structured data, indicating LLMs' limited ability in extracting knowledge from structured tables. The LLMs also perform less effectively when handling questions requiring the combination of both evidence types, reflecting their incapacity to integrate diverse structured evidence effectively.

**Analysis of Different Factual Questions Poses Challenges**    To assess the capabilities of LLMs in addressing various challenges, we partitioned each factual question within the structural task into six distinct challenges: 1) Entity disambiguation, 2) Other, 3) Multi-hop reasoning, 4) Combining tables and text, 5) Search terms not in claim, 6) Numerical reasoning, each centered around the most critical difficulty encountered during verification. Figure 3(c) illustrates GPT-3.5-Turbo's performance and data distribution across challenges. The extensive training and large-scale parameters enhance LLMs' performance in handling entity ambiguity. Longer reasoning chains and various forms of evidence challenge LLMs' factual abilities. When correct inference involves unmentioned entities, LLMs may lack necessary hints from factual questions, posing significant challenges. LLMs also exhibit deficiencies in precise numerical calculations due to the inherent hallucination phenomenon, resulting in subpar performance when numerical reasoning is needed for verification.

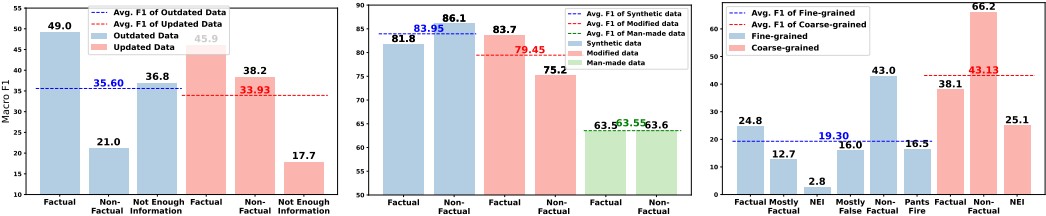

(a) Temporal Questions Verification   (b) Adversarial Attacks Resilience   (c) Label Granularity Variations

Figure 4: Results of GPT-3.5-Turbo in three different tasks under Few-shot CoT setting.

**Temporal Analysis**    As time progresses, the factuality of questions may undergo changes. This task encompasses such data, and we leverage this task to explore the ability of LLMs to adapt to factual changes. Figure 4(a) illustrates that GPT-3.5-Turbo exhibits a modest yet noticeable performance difference when dealing with outdated data as compared to updated data. This discrepancy arises from the fact that LLMs are pretrained on a corpus of text prior to a specific temporal point. Consequently, LLMs lack the capability to acquire real-time, up-to-date knowledge, rendering them unable to validate questions that hinge on the most recent information for accurate assessments.

**Adversarial Analysis**    To evaluate the robustness of LLMs to adversarial attacks, we divide the adversarial questions into three subsets: auto-generated questions from the corpus, manually modified synthesized questions yielding adversarial ones, and artificially created adversarial questions. Figure 3(b) presents the performance of GPT-3.5-Turbo on these three subsets. It is evident that following adversarial attacks, LLMs exhibit a substantial decrease in performance. Furthermore, factual questions that have undergone manual modifications or were artificially created prove to be more challenging compared to those that are automatically generated (Shen et al., 2023). This disparity could be attributed to the fact that automatically synthesized factual questions often contain explicit positive or negative words that hint at the outcome, and the exceptional comprehension abilities of LLMs enable them to accurately discern and provide the correct response in such cases.

**Label Granularity Analysis**    To assess the effect of different label granularities on LLMs' performance, we conducted a manual re-labeling of the real-world task questions. Per the settings of Misra (2022), besides labeling as "Factual", "Non-Factual", and "Not Enough Information", we also require them to annotate the dataset with six factual labels: "Factual", "Mostly Factual", "Mostly

False", "Non-Factual", "Pants-Fire", and "Not Enough Information". We also modified the prompt for GPT-3.5-Turbo for more intricate factual responses to test its competency with nuanced labels. Results in Figure 4(c) disclosed: 1) The results show that, in general, there is a significant decrease in performance (-23.83%) when transitioning from coarse-grained justification to fine-grained justification. With finer granularity, LLMs are not only required to assess the authenticity of each question but also to judiciously employ their knowledge base to precisely gauge the credibility of each factual questions. 2) When comparing the performance of coarse-grained labels with fine-grained labels, we observe significant drops in the three categories: "Factual" by 13.3%, "Non-Factual" by 23.2%, and "Not Enough Information" by 22.3%. This indicates that finer-grained labels introduce additional options that can potentially disrupt the original judgment of the LLMs. A potential remedy could be the aggregation of multiple judgments through voting (Wang et al., 2023a).

**Multilingual Task with Chinese and English Prompts**    To investigate the influence of prompts in different languages on LLMs, we extracted Chinese factual questions from the multilingual tasks to create a subset. We then evaluated the LLMs' performance when using both Chinese and English prompts, both of which are depicted in Appendix A.4. Table 4 illustrates the results, indicating that the LLMs perform better when using a Chinese prompt. This underscores the notion that employing prompts in the same language as the questions can enhance the transfer capabilities from English factual knowledge to other languages of LLMs.

| Language | English | Chinese |
|---|---|---|
| Factual | 41.7 | **55.5** |
| Non-Factual | 47.9 | **49.7** |
| NEI | **43.8** | 35.5 |
| Overall | 44.5 | **46.9** |

Table 4: Macro F1 over Chinese and English prompts.

Table 5: Results in different domains obtained on the Pinocchio-Lite using different prompts.

| Task | Multifaceted | | Structural | | Adversarial | | Temporal | | Real-World | | Domain Specific | | Multi-lingual | | Overall | |
|---|---|---|---|---|---|---|---|---|---|---|---|---|---|---|---|---|
| | Acc. | F1 | Acc. | F1 | Acc. | F1 | Acc. | F1 | Acc. | F1 | Acc. | F1 | Acc. | F1 | Acc. | F1 |
| 1 shot | 56.0 | 50.9 | 37.0 | 35.7 | 50.5 | 56.6 | 39.5 | 39.5 | 43.0 | 42.7 | 40.0 | 40.1 | 42.0 | 38.7 | 44.0 | 43.7 |
| 2 shots | 56.0 | **53.4** | 41.0 | 42.3 | 47.5 | 56.2 | 41.0 | 42.0 | 40.5 | 41.7 | **42.5** | **43.5** | 36.5 | 34.8 | 43.6 | 43.7 |
| 3 shot | 54.5 | 50.0 | 38.0 | 36.8 | 49.0 | 54.9 | 40.0 | 39.0 | 39.5 | 38.1 | 41.5 | 41.7 | 40.5 | 39.2 | 43.3 | 43.9 |
| 6 shots | 54.5 | 51.7 | 38.5 | 38.3 | 49.0 | 55.8 | 42.0 | 41.5 | 42.5 | 41.6 | 39.0 | 39.5 | 41.0 | 38.4 | 43.8 | 43.8 |
| 9 shots | **57.5** | 53.3 | 38.0 | 37.8 | 52.0 | 57.3 | 43.0 | 42.2 | 42.5 | 39.8 | 37.5 | 36.7 | 37.5 | 35.0 | 44.0 | 44.0 |
| 12 shots | 55.5 | 52.0 | 38.5 | 38.6 | 53.0 | 58.8 | **47.0** | **46.9** | **46.0** | **44.7** | 34.0 | 34.5 | 39.0 | 37.1 | 44.7 | 44.8 |
| Complex CoT | 51.0 | 50.2 | 38.5 | 35.0 | 37.5 | 47.2 | 39.0 | 39.0 | 39.5 | 36.8 | 36.0 | 35.7 | 38.5 | 31.7 | 40.0 | 39.7 |
| Self-Consistency | 55.5 | 51.2 | 43.0 | 42.6 | 49.5 | 54.8 | 43.0 | 41.6 | 43.0 | 41.9 | 42.0 | 42.4 | 39.5 | 36.8 | 45.1 | 45.0 |
| Self-Refinement | 55.0 | 52.1 | **44.5** | **44.0** | **53.5** | **59.2** | 42.5 | 42.2 | 41.5 | 40.3 | 42.0 | 43.4 | **43.0** | **39.9** | **46.0** | **46.2** |
| Declarative Claim | 52.0 | 51.1 | 39.0 | 35.1 | 45.5 | 49.3 | 40.5 | 40.7 | 40.0 | 37.9 | 41.0 | 40.6 | 38.5 | 36.3 | 42.3 | 41.6 |

**Prompt Strategy Analysis**    In prior research, various CoT methods have been employed to enhance the performance of LLMs. These methods include 1) augmenting the number of in-context learning examples, 2) implementing self-consistency mechanisms, which alleviates the hallucination phenomenon through majority voting after multiple judgments of LLMs (Wang et al., 2023a), 3) incorporating complex instances as demos to steer the cognitive processes of LLMs (Fu et al., 2022), and 4) employing self-refinement strategies, which refines LLMs' answers through continuous feedback of another LLM on responses to achieve better results (Madaan et al., 2023) and so forth. Additionally, we examined the influence of utilizing declarative claims as instances of in-context learning. We randomly sampled 200 factual questions from each task of the Pinocchio, totaling 1400 questions, to compose Pinocchio-Lite with the aim of speeding up the testing of different prompt strategies. The performance results of various CoT methods are presented in Table 5. To maintain fairness, three in-context learning examples are employed in the complex CoT, self-consistency, self-refinement, and declarative claim methods. Different types of CoT prompts are shown in Appendix A.4.

It is worth noting that 1) when the number of in-context learning examples is limited, the incremental improvement in performance is marginal upon increasing the number of examples. However, beyond a specific threshold, the addition of more examples gains more performance improvement. This could be due to the inability of LLMs to fully encapsulate the correct reasoning with fewer examples. 2) Concurrently, a fascinating observation is that the LLM's performance substantially deteriorates as the complexity of the CoT increases. This could stem from the difficulty LLMs have in extracting a generalized reasoning pattern from complex, multi-stage thinking processes with limited examples. 3) The self-consistency method markedly boosts performance by mitigating the hallucination issue in LLMs through consistency voting, enhancing their response accuracy. 4) In the self-refinement approach, the model might initially provide an incorrect response, but it can amend its mistakes through feedback and refine its answers. In the end, when no additional refinement is needed, the model often reaches the correct conclusion, achieving optimal performance. 5) Compared to the 3

shots method, the declarative claims method saw a 2.3% performance drop, illustrating that using questions as inputs better elicits factual knowledge than the original claim in the datasets.

## 5 RELATED WORK

**Factual Knowledge in Language Models**    Previous research shows that LLMs can retain and utilize factual knowledge, effectively acting as knowledge bases (Petroni et al., 2019; 2020; Heinzerling & Inui, 2021). This acquired factual knowledge in language models during pretraining can be advantageous for knowledge-intensive tasks like question answering and fact checking (Roberts et al., 2020; Yu et al., 2023a; Pan et al., 2023). To evaluate the factual knowledge stored in language models, Petroni et al. (2019) employed cloze tests consisting of triples and prompts specifically designed to simulate missing objects. Jiang et al. (2020b) explored the role of prompts in retrieving factual information from language models and devised improved prompts for probing. However, Elazar et al. (2021) demonstrated the unreliability of rank-based probing methods with paraphrased context, leading to inconsistent findings. Cao et al. (2021b) contended that biased prompts and leakage of golden answers often lead to overestimations of LLMs' knowledge storage capability. Our method is more in line with Kadavath et al. (2022) and Lin et al. (2022), employing self-evaluation by querying the models to assess response accuracy regarding factual knowledge.

More recent studies have directed their focus towards the detection of hallucinations—factually incorrect statements—in the responses generated by LLMs. For instance, the SelfCheckGPT (Manakul et al., 2023) uses a sampling method to detect inconsistencies in LLM responses, identifying hallucinated claims. Alternatively, FactScore (Min et al., 2023) approaches the challenge by deconstructing generations into atomic facts—concise statements—and assigning binary labels to assess their veracity. Furthermore, Chern et al. (2023) introduced a tool-enhanced framework for hallucination detection encompassing five core components: claim extraction, query formulation, tool-based querying, evidence gathering, and validation of consistency. However, these contributions primarily target the identification of factual inaccuracies in the models' output. In contrast, our benchmark is primarily designed to evaluate the breadth and depth of factual knowledge within LLMs.

**Benchmarks for Large Language Models**    The advent of LLMs has underscored the importance of exhaustive benchmarks for effective capability assessment. Presently, there are predominantly two types of existing benchmarks. One evaluates the general knowledge and reasoning capacities of LLMs, exemplified by the MMLU (Hendrycks et al., 2021a), a multi-choice benchmark that measures tasks from real-world tests and literature, spanning diverse subjects like elementary math, US history, computer science, and law. Moreover, benchmarks also exist for non-English languages (Huang et al., 2023) or in a bilingual context (Zhong et al., 2023). BIG-bench (Srivastava et al., 2022) is a collaborative benchmark examining LLMs' capabilities across 204 diverse tasks from various fields like linguistics, childhood development, software development, and more. HELM (Liang et al., 2022) employs 7 metrics over 42 tasks to assess LLMs, focusing on aspects from accuracy to robustness. Specific benchmarks like GSM8K (Cobbe et al., 2021) and MATH (Hendrycks et al., 2021a) target mathematical problem-solving, presenting elementary to competition-level problems. In program synthesis, HumanEval (Chen et al., 2021a) and MBPP (Austin et al., 2021) evaluate functional correctness through program synthesis from docstrings. Additional benchmarks address instruction following (Dubois et al., 2023), tool usage (Xu et al., 2023), and decision making (Liu et al., 2023). Our benchmark mainly focuses on factual knowledge, differing from ones like TruthfulQA (Lin et al., 2022), which specifically tests truthfulness in LLMs' generated responses, with questions structured to provoke imitative falsehoods over truthful answers.

## 6 CONCLUSION

In this work, our primary focus is the development of the Pinocchio benchmark, an extensive test bed encompassing 20,713 questions across seven varying complexity tasks, as a tool to investigate whether LLMs are capable of memorizing factual knowledge and reasoning on the basis of it. Upon applying the Pinocchio benchmark, we observe that various types of LLMs using different prompting strategies such as self-refine and self-consistency still have challenges in optimal performance on factual tasks. It is our hope that this novel benchmark will shed light on this area and act as a foundation for further improvements in LLMs' factual knowledge and reasoning abilities.

ACKNOWLEDGEMENT

This work is supported in part by NSF under grant III-2106758. Additionally, Junzhe Chen and Xiaochuan Li are supported by Beijing Natural Science Foundation under grant number QY23115 and QY23116.

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

# A APPENDIX

## A.1 ETHICAL STATEMENT

Pinocchio primarily serves to assess LLMs' responses to questions concerning factual knowledge. If a model performs effectively, it would be imprudent to infer that its reliability will uniformly translate to diverse task domains (even if some degree of transfer learning is anticipated). For instance, Pinocchio does not encompass long-form generation, such as news articles, or interactive settings, such as extended dialogues with adversarial entities. Furthermore, although the questions within Pinocchio parallel real-world inquiries, they originate not from a deployed system, thus posing a potential risk of over- or under-estimating the factuality of such a system.

We postulate that Pinocchio is unlikely to prove advantageous for those intending to fabricate deceptive models with malicious intent. To effectuate deception, a model must generate erroneous responses relatively infrequently, lest humans swiftly discern its unreliability. However, acquiring a low score on Pinocchio necessitates the provision of incorrect answers to virtually all questions. To be instrumental for malevolent purposes, a model must generate highly specific false statements, such as assertions concerning a maliciously targeted victim or a particular governmental policy. Yet, Pinocchio lacks coverage of highly specific subjects, offering instead a superficial overview of general factual topics.

While Wikipedia and some news websites are exemplary collaborative resources, they inherently contain inaccuracies and noise, akin to any encyclopedia or knowledge repository. Consequently, we advise users of Pinocchio against making absolute assertions about the validated claims and discourage its utilization for the development of truth-revealing models. We refrained from collecting participants' personal data in any form. Participants accessed our online tool exclusively using an identification number. Generated assertions must solely incorporate information deemed as general world knowledge or sourced from Wikipedia, thereby excluding any personally identifiable information or offensive content.

## A.2    THE DETAILED INTRODUCTION TO THE LLMs

For pretraining models, OPT (Zhang et al., 2022) is an open-sourced large causal language model which perform similar in performance to GPT-3 (Brown et al., 2020). BLOOM (Scao et al., 2022a) is an open-access multilingual large language model that is suitable for non-English facts. LLaMA (Touvron et al., 2023a) is probably the best open-weight foundation model so far that achieves the highest accuracy on various English benchmarks (e.g. MMLU (Hendrycks et al., 2021a)) within open-weight models. For instruction-tuned models, Alpaca (StanfordCRFM, 2023) is fine-tuned from the LLaMA model on 52K self-instructed demonstrations (Wang et al., 2023b). Alpaca behaves qualitatively similarly to OpenAI's Text-Davinci-003 on evaluation of single-turn instruction following. Vicuna is an open-source chatbot trained by fine-tuning LLaMA on user-shared conversations collected from ShareGPT (ShareGPT, 2023). Flan -T5 (Chung et al., 2022) is an enhanced version of T5 that has been instruction fine-tuned in a mixture of tasks. ChatGLM is an open bilingual language model based on the General Language Model (Zeng et al., 2023). ChatGLM is trained on Chinese and English corpus, supplemented by instruction tuning, feedback bootstrap, and reinforcement learning with human feedback (RLHF; Ouyang et al. 2022). ChatGPT (OpenAI, 2022) from OpenAI that has undergone pretraining, instruction tuning, and RLHF. ChatGPT has been observed to have impressive capabilities in various aspects favoring reasoning capabilities (Qin et al., 2023).

## A.3    TASK RESULTS

In this section, we present the results of all LLMs across different tasks under three different settings: Zero-shot w/o CoT, Zero-shot w/ CoT, and Few-shot w/o CoT.

Table 6: Results of different LLMs using Zero-shot w/o CoT prompts across different domains.

| Task | Multifaceted | | Structural | | Adversarial | | Temporal | | Real-World | | Domain Specific | | Multi-lingual | |
|------|------|------|------|------|------|------|------|------|------|------|------|------|------|------|
| | Acc. | F1 | Acc. | F1 | Acc. | F1 | Acc. | F1 | Acc. | F1 | Acc. | F1 | Acc. | F1 |
| OPT-6.7B | - | - | - | - | - | - | - | - | - | - | - | - | - | - |
| BLOOM-7B | 21.9 | 17.8 | 24.9 | 17.9 | 32.4 | 36.3 | 17.6 | 14.2 | 52.1 | 23.8 | 30.1 | 29.9 | 29.0 | 30.4 |
| LLaMA-7B | 30.7 | 28.8 | 38.3 | 29.3 | 30.8 | 35.6 | 37.9 | 26.0 | 35.1 | 32.4 | 27.1 | 29.1 | 13.9 | 17.2 |
| Alpaca-7B | 34.8 | 21.6 | 47.9 | 23.7 | 47.7 | 35.7 | **52.9** | 26.8 | 28.1 | 19.0 | 43.1 | 24.2 | 26.4 | 19.5 |
| Vicuna-7B | 38.6 | 35.4 | 19.4 | 16.8 | 50.8 | 53.9 | 37.9 | **42.0** | 29.8 | 30.1 | 33.6 | 30.4 | 34.8 | 34.4 |
| Vicuna-13B | 45.0 | 41.1 | 43.9 | 31.0 | 57.1 | 56.7 | 45.9 | 33.7 | 32.0 | 29.0 | 43.1 | 32.3 | 37.3 | 34.7 |
| ChatGLM-6B | 30.6 | 30.3 | 45.6 | 30.8 | 42.9 | 46.4 | 28.0 | 24.1 | 45.9 | 31.9 | 34.1 | 30.2 | 32.9 | 28.5 |
| Flan-T5-11B | 39.2 | 29.6 | 11.2 | 10.2 | 56.2 | 49.9 | 12.9 | 10.5 | 17.4 | 10.6 | 28.8 | 16.5 | 25.4 | 14.7 |
| Text-Davinci-002 | 44.7 | 38.4 | **49.2** | **37.8** | 57.2 | 56.1 | 36.2 | 27.8 | **53.2** | 32.7 | 31.3 | 30.1 | 42.2 | 32.5 |
| Text-Davinci-003 | 50.9 | 48.9 | 36.4 | 29.5 | 58.7 | 57.9 | 51.7 | 36.6 | 40.4 | 37.0 | 41.3 | 33.3 | 42.7 | 43.1 |
| GPT-3.5-Turbo | 53.2 | 50.1 | 43.1 | 35.8 | 62.3 | 61.8 | 43.4 | 35.9 | 46.1 | **42.1** | 42.5 | 35.6 | **45.0** | **45.7** |

Table 7: Results of different LLMs using Zero-shot w/ CoT prompts across different domains.

| Task | Multifaceted | | Structural | | Adversarial | | Temporal | | Real-World | | Domain Specific | | Multi-lingual | |
|------|------|------|------|------|------|------|------|------|------|------|------|------|------|------|
| | Acc. | F1 | Acc. | F1 | Acc. | F1 | Acc. | F1 | Acc. | F1 | Acc. | F1 | Acc. | F1 |
| OPT-6.7B | - | - | - | - | - | - | - | - | - | - | - | - | - | - |
| BLOOM-7B | 17.0 | 20.2 | 10.1 | 12.6 | 12.0 | 19.2 | 6.9 | 9.4 | 15.5 | 16.5 | 27.3 | 23.4 | 17.9 | 19.3 |
| LLaMA-7B | 20.3 | 23.5 | 29.5 | 26.4 | 18.3 | 26.2 | 25.7 | 26.3 | 22.9 | 24.9 | 20.0 | 23.0 | 12.2 | 16.9 |
| Alpaca-7B | 38.3 | 28.9 | 42.7 | 22.4 | 38.6 | 36.1 | 38.0 | 23.0 | 29.7 | 23.1 | 28.5 | 21.7 | 13.5 | 15.2 |
| Vicuna-7B | 29.4 | 35.8 | 45.7 | 31.6 | 4.4 | 8.3 | **49.0** | 36.6 | 15.1 | 19.6 | **47.4** | **39.6** | 37.9 | 33.9 |
| Vicuna-13B | 46.7 | 42.8 | 46.2 | 32.7 | 58.8 | 58.6 | 47.3 | 34.6 | 34.1 | 31.1 | 43.6 | 33.6 | 36.0 | 33.2 |
| ChatGLM-6B | 34.0 | 33.0 | 40.5 | 29.8 | 46.3 | 46.6 | 27.3 | 24.7 | 44.9 | 30.7 | 32.2 | 30.1 | 30.2 | 30.4 |
| Flan-T5-11B | 49.6 | 49.1 | 19.2 | 16.8 | 58.2 | 58.2 | 21.7 | 21.8 | 20.4 | 17.1 | 30.3 | 20.8 | 25.8 | 15.6 |
| Text-Davinci-002 | 47.2 | 40.1 | **51.7** | **38.0** | 59.9 | 58.2 | 37.2 | 30.8 | **52.7** | 34.4 | 29.9 | 30.3 | 42.5 | 36.6 |
| Text-Davinci-003 | 52.7 | 51.1 | 37.5 | 31.3 | 61.0 | 59.5 | 40.8 | 36.7 | 38.8 | 36.2 | 41.4 | 33.0 | 42.2 | 42.4 |
| GPT-3.5-Turbo | 53.3 | 52.1 | 43.1 | 35.5 | 59.8 | 61.6 | 42.2 | **37.7** | 44.8 | **43.3** | 41.4 | 36.0 | **43.4** | **45.3** |

Table 8: Results of different LLMs using Few-shot w/o CoT prompts across different domains.

| Task | Multifaceted | | Structural | | Adversarial | | Temporal | | Real-World | | Domain Specific | | Multi-lingual | |
|------|------|------|------|------|------|------|------|------|------|------|------|------|------|------|
| | Acc. | F1 | Acc. | F1 | Acc. | F1 | Acc. | F1 | Acc. | F1 | Acc. | F1 | Acc. | F1 |
| OPT-6.7B | 38.1 | 30.1 | 45.9 | 27.1 | 46.8 | 32.4 | 28.7 | 20.0 | 51.1 | 25.5 | 37.0 | 29.6 | - | - |
| BLOOM-7B | 32.7 | 22.5 | 8.8 | 9.0 | 43.5 | 32.6 | 23.8 | 21.1 | **53.3** | 31.4 | 29.3 | 28.4 | 22.3 | 19.3 |
| LLaMA-7B | 34.8 | 21.9 | 40.5 | 27.0 | 47.4 | 38.4 | 45.5 | 26.9 | 22.4 | 22.0 | 39.3 | 34.3 | 32.6 | 27.0 |
| Alpaca-7B | 34.9 | 25.4 | 48.0 | 22.6 | 43.4 | 32.5 | 48.0 | 25.8 | 24.0 | 19.4 | 42.6 | 27.0 | 21.8 | 17.4 |
| Vicuna-7B | 34.5 | 27.6 | 40.1 | 25.4 | 54.5 | 53.3 | 30.1 | 26.6 | 36.1 | 34.0 | 33.9 | 27.7 | 22.8 | 20.5 |
| Vicuna-13B | 47.9 | 42.5 | 48.9 | 31.4 | 54.7 | 53.1 | **53.4** | **38.6** | 39.7 | 35.2 | 47.4 | 34.9 | 37.7 | 36.8 |
| ChatGLM-6B | 37.9 | 32.9 | 44.6 | 35.4 | 52.2 | 46.8 | 44.9 | 35.4 | 38.0 | 33.9 | 41.6 | 38.0 | 34.5 | 33.8 |
| Flan-T5-11B | 42.3 | 35.0 | 12.4 | 11.7 | 57.7 | 53.6 | 15.1 | 13.0 | 17.7 | 11.4 | 29.7 | 19.4 | 24.9 | 13.6 |
| Text-Davinci-002 | 45.4 | 41.2 | **51.4** | **38.4** | 61.7 | 61.8 | 37.0 | 31.3 | 52.0 | 38.6 | 33.0 | 32.6 | 42.5 | 40.0 |
| Text-Davinci-003 | **59.6** | 43.4 | 48.1 | 33.7 | 62.0 | **61.8** | 46.4 | 36.3 | 50.6 | 43.0 | 41.7 | 36.3 | **44.2** | **44.4** |
| GPT-3.5-Turbo | 52.1 | 48.4 | 42.5 | 35.4 | 61.2 | 61.1 | 43.7 | 36.2 | 48.9 | **43.2** | 42.0 | 35.6 | 42.8 | 43.0 |

## A.4 PROMPT STRATEGY

In this section, we provide the comprehensive versions of all the prompts utilized in both the main experiments and the subsequent analysis. We engaged native Chinese annotators to rephrase the English prompts while maintaining their semantic integrity, thus yielding Chinese prompts.

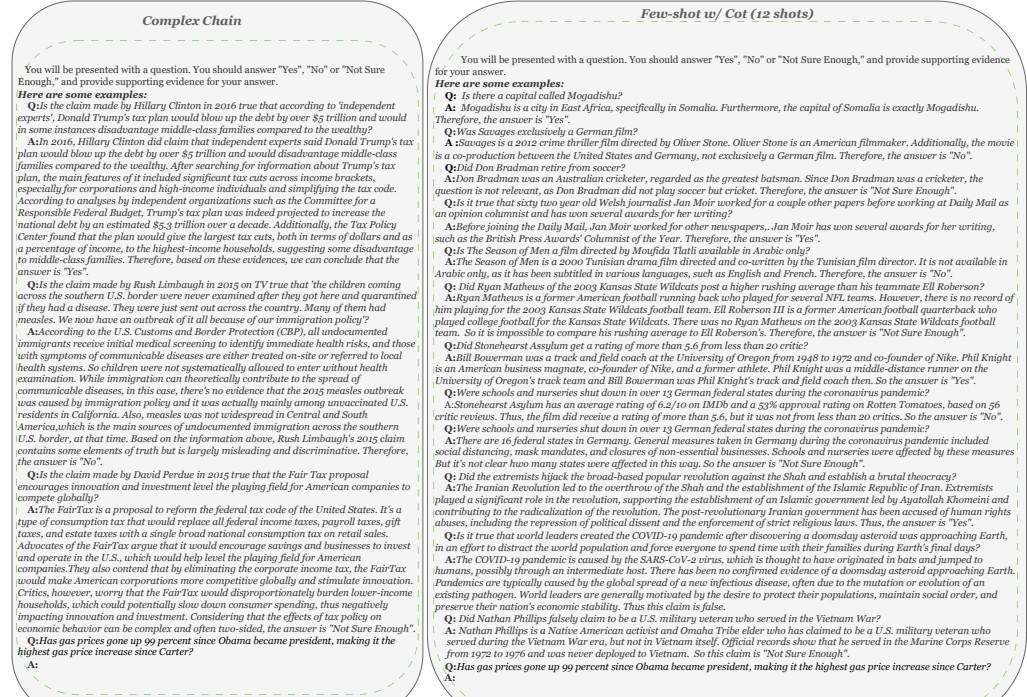

Figure 5: Prompts of four different settings.

Figure 6: Prompts of complex chain and Few-shot CoT with 12 shots method.

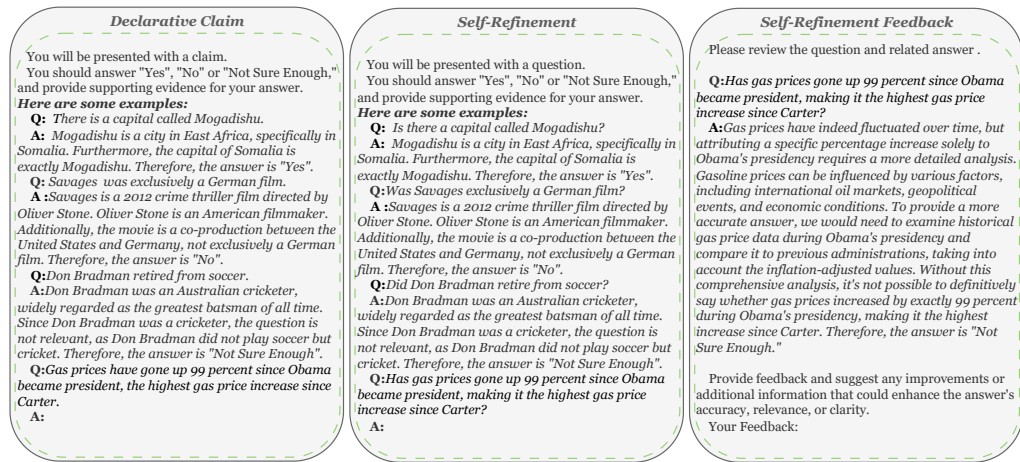

Figure 7: Prompts of self-refinement and declarative claim method.

## A.5 CASE STUDY

We have introduced an additional scenario for investigation, which occurs frequently in the output generated by the zero-shot prompt method. We conducted an experiment involving three models: OPT, ChatGLM, and GPT-3.5-Turbo. These models are presented with the same set of questions, and their responses are shown in Figure 8. It is noteworthy that the OPT model, in both questions, reiterated the question itself without providing the corresponding answer. It is essential to mention that the actual output of the OPT model repeats the problem until it reaches the maximum output length (controlled by the "max_length" parameter), and we truncated the repeated portion.

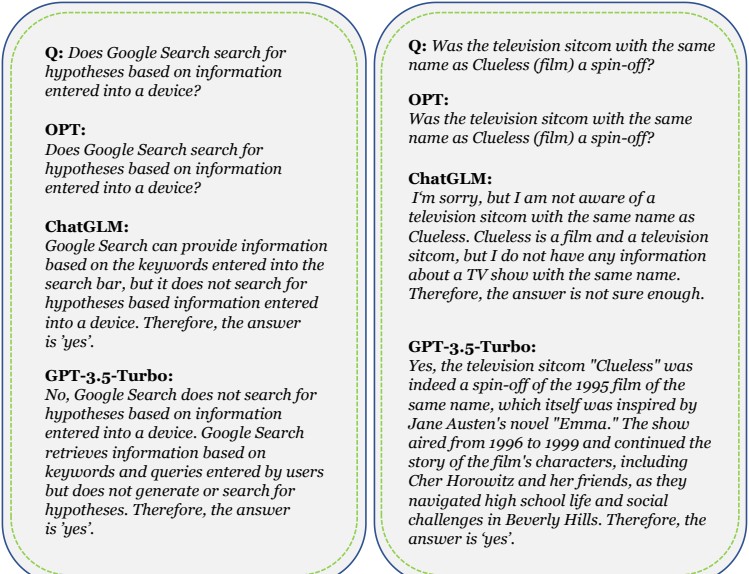

Figure 8: Answers to the same question from different LLMs in the zero-shot setting.

The OPT model even declined to generate any content when presented with the zero-shot prompt, resulting in a significant number of empty responses in the statistical results. In the first question, both ChatGLM and GPT-3.5-Turbo provided correct answers. However, in the second question, when faced with more detailed information inquiries, ChatGLM failed to produce a correct response, while GPT-3.5-Turbo demonstrated proficient reasoning and provided accurate answers. Increasing

the number of shots in a few-shot learning setting also did not yield a substantial improvement in performance.

## A.6 FACT-CHECKING DATASETS

In this section, we first undertook a comprehensive survey of numerous existing Fact-Checking datasets, the summary of which is presented in Table 13. Our analysis focused on identifying the various challenges these datasets aim to address regarding factuality. We found that the challenges tackled by these datasets generally align with the seven aspects we have listed in our dataset. These aspects either appear individually or in combination across the surveyed datasets, indicating their relevance and importance in the field of fact-checking. This realization led us to intentionally design our evaluation framework around these seven specific challenges, ensuring that our benchmark is not only comprehensive but also directly addresses the core difficulties encountered in current fact-checking tasks.

Table 9: Domain Distribution of Various Fact-Checking Datasets.

| Dataset | Multifaceted | Structural | Adversarial | Temporal | Real-World | Domain-Specific | Multi-Lingual |
|---|---|---|---|---|---|---|---|
| COVID-19 Disinfo (Alam et al., 2021) | ✓ | | ✓ | | | | ✓ |
| SPICED (Wright et al., 2022) | ✓ | | ✓ | ✓ | | ✓ | |
| EMU (Da et al., 2021) | ✓ | | | | | | |
| NeuralNews (Tan et al., 2020) | ✓ | | | | | | |
| Propa-News (Huang et al., 2022) | ✓ | | | | | | |
| HOVER (Jiang et al., 2020a) | ✓ | | | | | | |
| ParsFEVER (Zarharan et al., 2021) | ✓ | | | | | | |
| MultiFC (Augenstein et al., 2019) | ✓ | | | | ✓ | | |
| Fact-KG (Kim et al., 2023a) | ✓ | ✓ | | | | | |
| NewsCLIPpings (Luo et al., 2021) | | ✓ | | | | ✓ | |
| Semeval 2021 Task9 (Wang et al., 2021) | | ✓ | | | | | |
| Infotabs (Gupta et al., 2020) | | ✓ | | | | | |
| TabFact (Chen et al., 2019) | | ✓ | | | | | |
| InfoSurgeon (Fung et al., 2021) | ✓ | | ✓ | | | | |
| DeSePtion (Hidey et al., 2020) | ✓ | | ✓ | | | | |
| RumorEval19 (Gorrell et al., 2019) | | | ✓ | | ✓ | | |
| AdverBenc (Flores & Hao, 2022)h | | | ✓ | | ✓ | | |
| Fakeedit (Nakamura et al., 2019) | | | ✓ | | | ✓ | |
| Claimde-Comp (Chen et al., 2022) | | | ✓ | | | ✓ | |
| AVeriTec (Schlichtkrull et al., 2023) | | | | ✓ | ✓ | | |
| VoynaSlov (Park et al., 2022) | | | | ✓ | | | ✓ |
| WatClaimCheck (Khan et al., 2022) | | | | ✓ | ✓ | | |
| MuMiN (Nielsen & McConville, 2022) | | ✓ | | ✓ | ✓ | | ✓ |
| MR$^2$ (Hu et al., 2023) | | | | | ✓ | | ✓ |
| FakeSV (Qi et al., 2023) | | | | | ✓ | | ✓ |
| Weibo20 (Rao et al., 2021) | | | | | ✓ | | ✓ |
| Rumor Stance (Lillie et al., 2019) | | | | | ✓ | | ✓ |
| Veritas (Azevedo et al., 2021) | | | | | ✓ | | |
| LIAR (Wang, 2017) | | | | | ✓ | ✓ | |
| FakeNewsNet (Shu et al., 2020) | | | | | ✓ | ✓ | |
| ClaimBuster (Arslan et al., 2020) | | | | | ✓ | ✓ | |
| CURT (Sundriyal et al., 2022) | | | | | ✓ | | |
| Health-VER (Sarrouti et al., 2021) | | | | | ✓ | ✓ | |
| Covid-Fact (Saakyan et al., 2021) | ✓ | | | | | ✓ | |
| CoVERT (Mohr et al., 2022) | ✓ | | | | | ✓ | |
| Answer-Fact (Zhang et al., 2020) | ✓ | | | | | ✓ | |
| SciTweets (Hafid et al., 2022) | | | | | | ✓ | ✓ |
| Dial-Fact (Gupta et al., 2021) | | | | | | ✓ | |
| CHEF (Hu et al., 2022a) | | | | | ✓ | | ✓ |
| DanFEVER (Nørregaard & Derczynski, 2021) | ✓ | | | | | | ✓ |
| Stanceosaurus (Zheng et al., 2022) | ✓ | | | | | | ✓ |
| Climate-FEVER (Diggelmann et al., 2020) | | | | | | | ✓ |
| FakeCovid (Shahi & Nandini, 2020) | | | | | | ✓ | ✓ |

## A.7 EVIDENCE ANALYSIS

In this section, we manually analyze the inference process of the LLMs by utilizing their outputs under the few shots with CoT setting. We manually inspected a sample of 100 instances where LLMs provided reasoning outputs. The key observations from our analysis include:

1. Correct Reasoning Leading to Correct Answers: In 38% of the instances, the LLMs demonstrated a correct reasoning process, which corresponded with the accurate answers they provided. As shown in Figure 9 (a), the LLM accurately processed a query regarding the central parity rate of the RMB against the US dollar on November 30, 2020, resulting in a correct answer classified as "Factual".

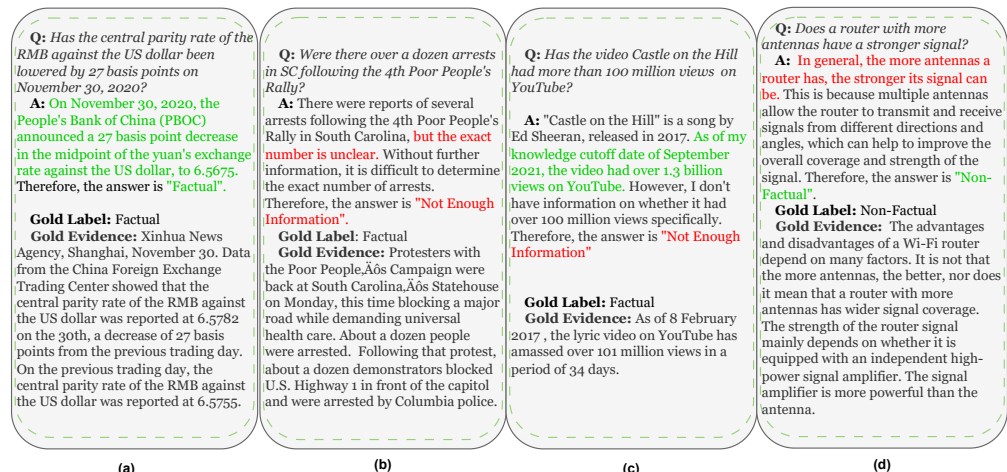

Figure 9: Resoning process from LLMs in the few-shots w/ CoT setting.

2. Flawed Reasoning Leading to I Errors: 51% of the instances exhibited incorrect reasoning processes, which inevitably led to incorrect conclusions. As an example, consider Figure 9 (b): The question posed was whether there were over a dozen arrests in South Carolina following the 4th Poor People's Rally. The LLM responded by stating there were reports of several arrests, but it could not ascertain the exact number, leading to a conclusion of "Not Enough Information." However, the 'Gold Evidence' clearly stated that about a dozen people were arrested following the demonstration, indicating that the correct label should have been "Factual." This instance underscores a scenario where the LLM might not possess the specific numerical details present in the gold evidence, leading to an incomplete and therefore inaccurate conclusion.

3. Discrepancies in Reasoning and Conclusions: Interestingly, in our analysis, we identified cases where the reasoning process did not align with the final conclusion. Specifically, there were instances, comprising 4% of our sample, where the LLMs reached a correct answer despite flawed reasoning. For example, in Figure 9 (c), the question was about whether Ed Sheeran's "Castle on the Hill" had more than 100 million views on YouTube. The LLM correctly concluded that it had over 1.3 billion views by September 2021, but stated "Not Enough Information" for the specific query of 100 million views, despite the fact being factual as per the gold evidence. Conversely, in 6% of the cases, the LLMs displayed sound reasoning but still arrived at an incorrect conclusion. A notable example of this is seen in Figure 9 (d), where the question asked whether a router with more antennas has a stronger signal. The LLM reasoned that a greater number of antennas would generally lead to a stronger signal due to enhanced multi-directional transmission and reception capabilities. Despite this logical reasoning, the LLM concluded the answer as "Non-Factual", which contradicts the reasoning provided.

4. Internal Reasoning Contradictions: We also found an example where the reasoning process of the LLM was contradictory. The question is "*Does a deficiency of vitamin B6 decrease blood levels of homocysteine?*" and LLM answers: "*Vitamin B6 is involved in the metabolism of homocysteine, which is an amino acid that is potentially harmful to blood vessels when present in high levels. A deficiency of vitamin B6 can lead to an accumulation of homocysteine in the blood. Therefore, a deficiency of vitamin B6 can decrease blood levels of homocysteine. The answer is "Factual"."* Initially, the LLM correctly states that a deficiency in vitamin B6 can lead to an accumulation of homocysteine. However, it concludes with a contradictory statement, suggesting that a deficiency in vitamin B6 can decrease homocysteine levels.

## A.8 BASELINE ANALYSIS

Following Chen & Durrett (2019), we developed the same "no context" baseline to investigate the spurious correlations between questions and labels in our dataset. The results are shown in Table 10:

Table 10: The performance of different models across Pinocchio.

| Datasets | Accuracy(%) |
|----------|-------------|
| No Context | 28.3 |
| LLaMA-7B | 31.6 |
| Alpaca-7B | 37.8 |
| Vicuna-13B | 45.2 |
| GPT-3.5 | 47.0 |

Our experimental findings show that our dataset does not exhibit the same level of vulnerability to the exploitation of question-label correlations as observed by Chen & Durrett (2019) in the WikiHop dataset. With performance improvements of 16.9 points by Vicuna-13B and 18.7 by GPT-3.5 over the "no context" baseline, our results offer compelling evidence that our dataset is more resilient to such biases, contrary to the reported susceptibilities within WikiHop.

We extended our analysis to include a direct comparison with several established multiple-choice question-answering benchmarks, such as the WikiHop mentioned above, as well as with other prevalent benchmarks like TruthfulQA and ARC utilized in evaluating LLMs. The performance of the "no context" baseline across these benchmarks is displayed in the Table 11:

Table 11: The performance of the "no context" baseline across these benchmarks.

| Datasets | Accuracy(%) |
|----------|-------------|
| WikiHop  (Welbl et al., 2018b) | 59.7 |
| TruthfulQA  (Lin et al., 2021) | 34.5 |
| ARC  (Clark et al., 2018b) | 33.2 |
| Ours | 28.3 |

Evidently, our proposed dataset presented the most challenge to the "no context" baseline, marking the lowest performance compared to other datasets. The notable performance on WikiHop, with a "no context" baseline score of 59.7%, underscores the presence of spurious correlations that facilitate gaming that dataset. On the contrary, the lower baseline performances on TruthfulQA and ARC suggest that such issues are less prevalent. Our dataset, therefore, not only stands out as the least prone "to be gamed" but also underscores its robustness and the high level of rigor needed to tackle it effectively.

## A.9    PEER-TO-PEER ANALYSIS

The comparisons between LLaMA and its instruction-tuned versions, Alpaca and Vicuna, can be found in Table 2. Furthermore, we have conducted extra tests under the few-shots with CoT setting for T5-11B vs. Flan-T5-11B and BLOOM-6.7B vs. BLOOMz-6.7B as shown in Table 12. For T5, the accuracy was 18.6%, and the Macro F1 was 25.2%. In contrast, as shown in Table 2, Flan-T5 achieved an accuracy of 38.4% and a Macro F1 of 38.4%. Similarly, BLOOM's performance was at an accuracy of 6.6% and Macro F1 of 12.2%, whereas BLOOMz showed a marked improvement with an accuracy of 27.5% and a Macro F1 of 27.7%. These peer-by-peer comparisons reveal that, with few exceptions (e.g., LLaMA vs. Alpaca in terms of Macro F1), models that underwent instruction tuning generally outperform their backbone counterparts, achieving an average improvement of 11.3%.

## A.10    RELATED WORK: QUESTION ANSWERING DATASETS

In this section, we offer a thorough examination of existing question-answering initiatives as they pertain to the seven key dimensions that form the core of our benchmark. These dimensions are multifaceted, structural, adversarial, temporal, real-world, and multilingual.

As detailed in Table 13, we present a comprehensive overview of notable datasets within the realm of question-answering. We categorize these datasets based on several criteria to illuminate their

Table 12: Peer-to-peer comparison between the instruction-tuned models and their backbones.

| Models | Accuracy(%) | Macro F1(%) |
|---|---|---|
| LLaMA-7B | 35.3 | 31.4 |
| Alpaca-7B | 39.4 | 26.2 |
| Vicuna-7B | 48.5 | 40.6 |
| T5-11B | 18.6 | 25.2 |
| Flan-T5-11B | 38.4 | 38.4 |
| Bloom-6.7B | 6.6 | 12.2 |
| Bloomz-6.7B | 27.5 | 27.7 |

distinctive challenges and characteristics. First, we identify the "Type" of challenge each dataset presents. Next, "Source" provides the origins of the questions. "Retrieval" indicates the necessity of sourcing external knowledge, such as documents, to formulate an answer. When it comes to the "Answer types", datasets may require various forms of responses ranging from multiple-choice options (A, B, C, etc.), specific text spans (e.g., an entity or a phrase), to Boolean (yes or no) and free-form answers that allow for the generated text of any length. "Domain" captures the field to which the questions belong, encompassing areas like science, biography, or geography.

Interestingly, beyond these seven axes, there exist other datasets that probe the knowledge and reasoning capabilities of large language models (LLMs) from different perspectives. For instance, research centered around knowledge updating, particularly focusing on entities, has been conducted. Onoe et al. (2022) delve into the ability of LLMs to make inferences about newly emerged entities that were not part of the LLMs' pretraining data. Building on this, Onoe et al. (2023) investigated the extent to which LLMs can integrate descriptions of new entities. On the other hand, Peng et al. (2022) have assessed LLMs' conceptual knowledge by crafting three distinct tasks that test whether LLMs are capable of categorizing entities based on conceptual similarities.

**Multifaceted** Existing efforts in question answering that relate to the multi-faceted nature of our dataset predominantly encompass multi-hop reasoning datasets. These datasets necessitate models to synthesize multiple information snippets to formulate an answer. For instance, WikiHop Welbl et al. (2018a) constructs a bipartite graph from a knowledge base populated with relational triplets. This graph undergoes a breadth-first traversal to yield valid multi-hop reasoning chains. Similarly, HotpotQA (Yang et al., 2018) narrows its focus to 2-hop questions derived from the initial sections of English Wikipedia documents. The selection of two passages to form a reasoning chain is predicated upon one of two conditions: either a hyperlink connects the first document to the second, or the associated entities belong to an identical category. Moving onto a broader spectrum, MultiRC (Khashabi et al., 2018) introduces multi-domain multi-hop questions. It compiles documents from various domains and a multitude of datasets, where the different contexts are all embedded within the same textual passage. As opposed to the questions themselves providing explicit decompositional cues, StrategyQA (Geva et al., 2021) conceals the necessary reasoning steps within the question itself. These steps must be astutely deduced using strategic inference.

Additionally, several datasets intertwine multi-hop reasoning with further complexities. Open-BookQA (Mihaylov et al., 2018) offers a specialized challenge, combining question answering techniques with a compendium of scientific facts to assess knowledge in the scientific domain, supplemented by a broader base of common understanding. In a vein similar to OpenBookQA, QASC (Khot et al., 2020) also revolves around two-hop question answering with a foundation in scientific facts; its methodology for reasoning chain generation closely resembles that of OpenBookQA. Furthermore, datasets like HybridQA (Chen et al., 2020) and OTTQA (Chen et al., 2021b) venture into multi-hop reasoning across both tabular and textual data sources. Vu et al. (2023) introduces FreshQA, incorporating questions that demand multi-hop reasoning where answers may shift over time, as well as tackling premises that are fundamentally flawed.

**Structural** Structured and semi-structured knowledge are known as unambiguous and compositional. Traditional question answering datasets predominantly cater to uniform types of information, either focusing exclusively on textual data or relying solely on knowledge bases and tables (Berant et al., 2013; Talmor & Berant, 2018). This approach, however, overlooks the complexity of human

knowledge which is inherently diverse and spread across varied formats. Relying solely on homogenous sources may result in limited scope and inadequate coverage of information. Addressing this gap, Chen et al. (2020) proposed HybridQA, a dataset that necessitates reasoning over a blend of heterogeneous information sources. In HybridQA, each question necessitates the integration of information from a Wikipedia table and assorted text corpora tied to the entities mentioned within the table, thereby combining tabular and textual data. Parallel initiatives targeting niche fields also emerged, with Li et al. (2021) focusing on geographical data and Zhu et al. (2021) on financial information. These domain-specific endeavors highlight the growing interest in incorporating structural knowledge. Departing from the provision of pre-selected tables and textual passages, OTTQA (Chen et al., 2021a) and NQ-table (Herzig et al., 2021) propel the question-answering challenge into the open-domain setting. Here, the retrieval of pertinent tables and text from comprehensive sources like Wikipedia becomes an integral part of the task. Our structural task aligns more closely with the objectives of OTTQA and NQ-table, where LLMs are tasked with performing advanced multi-hop inference. This entails navigating through a combination of both structural and unstructured factual knowledge to deduce accurate answers, reflecting a more realistic and complex information processing challenge akin to the ways humans interact with a variety of knowledge types to make informed decisions.

**Adversarial**    Machine learning models have a known susceptibility to adversarial examples—inputs that have been intentionally modified to cause a model to make a mistake. A notable instance within the realm of question answering tasks is the presence of questions based on dubious assumptions, which are typically classified as unanswerable questions (Rajpurkar et al., 2018; Kwiatkowski et al., 2019; Asai & Choi, 2021). More recently, Kim et al. (2021) critiqued the practice of lumping questions with dubious assumptions into the 'unanswerable' category as inadequate. They advocated for employing presuppositions within explanations as a means to more effectively determine their unanswerability. Additionally, their work demonstrates the complexity of verifying assumptions, proving it to be a formidable challenge even in closed-book environments. Building upon this, Kim et al. (2023b) expanded the investigation into open-domain contexts, confirming the inherent difficulties associated with QA that involve problematic assumptions. They discovered that, even when the hurdle of recognizing assumptions is eliminated, the task of factual verification remains unsolved—though, it should be noted, recent enhancements in LLMs have indeed contributed to some progress in verification capabilities. In a related vein, Yu et al. (2023b) presented a new open-domain QA dataset that features a natural distribution of failures due to presuppositions. Their research reveals that the challenges in handling questions with questionable assumptions are consistent, irrespective of the different sources from which the questions are derived, which include search engine prompts as well as Reddit inquiries. This body of work indicates that while strides have been made in addressing some aspects of QA tasks, the nuanced issue of dealing with questionable assumptions persists across various settings and requires further exploration.

**Temporal**    Understanding the temporal evolution of information is a significant area of interest in the field of question answering. Initial research, such as TempQuestions (Jia et al., 2018), investigated temporal aspects of questions that incorporated time specifiers within knowledge bases. Subsequent studies have shifted their focus toward apprehending the nuances of temporal progression in natural language texts. For example, Chen et al. (2021c) introduced TimeQA, a resource constructed by extracting and compiling evolving facts from WikiData alongside corresponding Wikipedia passages, resulting in a dataset of 20,000 timestamped question-answer pairs. Moreover, Zhang & Choi (2021) presented SituatedQA, which includes 9,000 realistically formulated questions from pre-existing open-domain QA datasets, each complemented with temporal contexts, such as specific timestamps. StreamingQA (Liska et al., 2022) is another relevant contribution that encompasses a blend of machine-generated and human-authored questions—altogether totaling 146,000 entries—designed to be answerable using a repository of timestamped news articles. In the same vein, the dynamic RealTimeQA benchmark (Kasai et al., 2022b) poses a challenge for models by offering 30 multiple-choice questions based on recent events curated from news websites, thereby testing their ability to handle fresh content. Adding to these advancements, FreshQA (Vu et al., 2023) brings a new dimension to the table with a static compilation of human-curated open-ended questions. The uniqueness of FreshQA lies in the evolving nature of its answers, which are subject to change in response to ongoing world developments, providing a generative assessment for time-sensitive question answering. This body of work collectively underscores the complexity and dynamism inherent in temporal question answering research.

**Domain-Specific** While there have been successful developments in question-answering within broad domains, specialized domains such as science and biomedicine remain relatively underexplored and present unique challenges. The limited availability of domain-specific datasets, coupled with the need for an in-depth understanding of specialized knowledge to match that of human experts, marks these areas as fertile ground for ongoing research. In the scientific domain, existing datasets necessitate the use of varied reasoning methods tailored to each specific question (Clark et al., 2018a). For instance, the OpenBookQA dataset (Mihaylov et al., 2018) presents multiple-choice questions that are generated based on a core book of fundamental science facts. Similarly, the QASC dataset (Khot et al., 2020) offers multiple-choice questions on science topics appropriate for elementary and middle school levels, emphasizing the combination of facts. QASC is unique in that it intentionally includes pairs of facts that, according to evaluations by crowd workers, provide enough information to deduce the answer to each question. Shifting the focus to the biomedical field, a range of new datasets have emerged to support question-answering tasks that hinge on domain-specific expertise. These include datasets such as HealthQA (Zhu et al., 2019), MASH-QA (Zhu et al., 2020a), and MedMCQA (Pal et al., 2022), which have been introduced to bolster research in medical question-answering applications. These datasets serve as valuable resources to address the nuanced queries that arise within the complex terrain of biomedical knowledge.

**Multi-Lingual** Recent effort has been made to create non-English QA datasets to overcome the data scarcity in non-English languages, typically including one or two languages. These include DuReader (He et al., 2018) in Chinese, French/Japanese evaluation sets for SQuAD created via translation (Asai et al., 2018), a semi-automatic Italian translation of SQuAD (Croce et al., 2019), ARCD—an Arabic reading comprehension dataset (Mozannar et al., 2019), a Hindi-English parallel dataset in a SQuAD-like setting (Gupta et al., 2018), and a Chinese–English dataset focused on visual QA (Gao et al., 2015). Recent datasets cover more languages, such as XQuAD (Artetxe et al., 2020) and MLQA (Lewis et al., 2020), which are examples of SQuAD-style extractive datasets, employing human translators to create parallel examples. MLQA and XQuAD ensure that all answers are answerable, and derive answers from provided documents. Instead of extractive answers, Hardalov et al. (2020) introduced EXAMS, a multilingual multiple-choice QA from school exams. TyDiQA (Clark et al., 2020) and MKQA (Longpre et al., 2021), focus on typological diversity in its wide language selection. While TyDiQA offers a more natural distribution of questions, its annotations are based on the retrieval system used by the authors (Google search); hence their answers are actually start and end indices for spans of text within a given passage. Xor QA (Asai et al., 2021) explores cross-lingual subtasks by re-annotating TyDiQA examples, sourcing answers from English documents, and translating them back to the target language. While state-of-the-art models have matched or surpassed human performance in general-purpose monolingual benchmarks, current methods still fall short of human performance on multilingual benchmarks, despite recent gains. Multilingual question answering consequently is at the frontier of such cross-lingual generalization.

Table 13: A comprehensive comparison of question answering datasets.

| Dataset | Type | Source | Retrieval | Answer Type | Domain |
|---|---|---|---|---|---|
| WikiHop (Welbl et al., 2018a) | Multifaceted | WikiData | ✗ | Multiple Choice | General |
| HotpotQA (Yang et al., 2018) | Multifaceted | Wikipedia | ✓ | Span | General |
| MultiRC (Khashabi et al., 2018) | Multifaceted | Multiple | ✗ | Multiple Choice | General |
| StrategyQA (Geva et al., 2021) | Multifaceted | Wikipedia | ✓ | Boolean | General |
| OpenBookQA (Mihaylov et al., 2018) | Multifaceted/Domain-Specific | WorldTree | ✓ | Multiple Choice | Science |
| QASC (Khot et al., 2020) | Multifaceted/Domain-Specific | Wikipedia | ✓ | Multiple Choice | Science |
| NQ-tables (Herzig et al., 2021) | Structural | Google Queries | ✓ | Span | General |
| TAT-QA (Zhu et al., 2021) | Structural/Domain-Specific | Wikipedia | ✗ | Span | Finance |
| TSQA (Li et al., 2021) | Structural/Domain-Specific | Exam | ✗ | Multiple Choice | Geography |
| HybridQA (Chen et al., 2020) | Structural/Multifaceted | Wikipedia | ✗ | Span | General |
| OTTQA (Chen et al., 2021b) | Structural/Multifaceted | Wikipedia | ✓ | Multiple Choice | General |
| $(QA)^2$ (Kim et al., 2023b) | Adversarial | Google Queries | ✗ | Free-form | General |
| CREPE (Yu et al., 2023b) | Adversarial/Real-World | Reddit | ✗ | Free-form/Boolean | General |
| TempQuestions (Jia et al., 2018) | Temporal | Datasets | ✗ | Free-form/Boolean | General |
| TimeQA (Chen et al., 2021c) | Temporal | WikiData | ✗ | Span | General |
| SituatedQA (Zhang & Choi, 2021) | Temporal | Datasets | ✓ | Span | Geography |
| RealTimeQA (Kasai et al., 2022a) | Temporal | News | ✓ | Multiple-Choice | General |
| StreamingQA (Liska et al., 2022) | Temporal | News | ✓ | Free-form | General |
| FreshQA (Vu et al., 2023) | Temporal/Multifaceted | Manual | ✓ | Free-form | General |
| MSMarco (Nguyen et al., 2016) | Real-World | Bing Queries | ✓ | Free-form | General |
| SearchQA (Dunn et al., 2017) | Real-World | Google Queries | ✓ | Span | General |
| TriviaQA (Joshi et al., 2017) | Real-World | Forum | ✓ | Span | General |
| DuReader (He et al., 2018) | Real-World | Baidu Queries | ✓ | Free-form | General |
| NQ (Kwiatkowski et al., 2019) | Real-World | Google Queries | ✓ | Free-form | General |
| ELI5 (Fan et al., 2019) | Real-World | Reddit | ✓ | Free-form | General |
| ARC (Clark et al., 2018a) | Domain-Specific/Multifaceted | Search Queries | ✓ | Multiple Choice | Science |
| QASPER (Dasigi et al., 2021) | Domain-Specific | Papers | ✓ | Span | Science |
| ScienceQA (Lu et al., 2022) | Domain-Specific | Exams | ✗ | Multiple Choice | Science |
| HealthQA (Zhu et al., 2019) | Domain-Specific | Patient | ✗ | Free-form | BioMed |
| MedMCQA (Pal et al., 2022) | Domain-Specific | Exams | ✗ | Multiple Choice | BioMed |
| MASH-QA (Zhu et al., 2020b) | Domain-Specific | WebMD | ✗ | Free-form | BioMed |
| XQuAD (Artetxe et al., 2020) | Multilingual | SQuAD | ✗ | Span | General |
| MLQA (Lewis et al., 2020) | Multilingual | Wikipedia | ✗ | Span | General |
| EXAMS (Hardalov et al., 2020) | Multilingual | Exam | ✗ | Multiple Choice | General |
| TydiQA (Clark et al., 2020) | Multilingual/Real-World | NQ | ✗ | Span | General |
| MKQA (Longpre et al., 2021) | Multilingual/Real-World | NQ | ✗ | Multiple | General |
| XOR QA (Asai et al., 2021) | Multilingual/Real-World | TydiQA | ✓ | Span | General |

