# OpenReview forum: "Towards Understanding Factual Knowledge of Large Language Models"
_ICLR.cc/2024/Conference — ICLR 2024 spotlight_

### Official Review · Reviewer_SmAT · 2023-10-26

**Soundness:** 3 good
**Presentation:** 3 good
**Contribution:** 3 good
**Rating:** 8
**Confidence:** 4

**Summary:**

This work presented a benchmark containing 20K diverse factual questions that span different sources, timelines, domains, regions, and languages. This work also thoroughly investigated different sizes and types of LLMs on this benchmark and reported their findings.

**Strengths:**

This work presented a comprehensive benchmark for testing LLMs' factual and commonsense knowledge.

This work annotated 20K factual questions from seven diverse tasks, which provides a good testbed for the community.

The analyses presented are solid and insightful.

**Weaknesses:**

~Actually, I am satisfied the most part of this paper and willing to raise my score if the author can cover the following points:~

Some points in the main results could be verified through better experiment settings, such as the point that "LLMs without instruction tuning underperform those with instruction tuning by 16.0%". It's better to conduct a peer-to-peer comparison since the instruction tuning is not the only factor of the two groups. For example, we can compare LLaMA with Alpaca and Vicuna, OPT vs. OPT-IML, T5 vs. Flan-T5, BLOOM vs. BLOOMz. I am not saying that we need to compare all of them, but the comparison could be made between the instruction tuning model and its backbone.

For the "Using the CoT method, we observed a relative boost in performance in LLMs subjected to
instruction tuning and RLHF", the tested models may not have strong CoT ability. It's good to see LLaMA-2 or GPT-4 and other recent models that claim to have strong reasoning abilities. Since CoT is not a generic ability that every LLM has.

Besides, you can include an additional metric to report the cases that LLMs failed to generate meaningful answers. Since LLM may refuse to answer according to security or other customized policies, treating all unanswered questions as wrong predictions may be unfair.

Sec-4.2 could be improved, especially the discussion around temporal analysis, " GPT-3.5-Turbo exhibits superior performance when dealing with outdated data as compared to updated data.", I think the numbers are close, and it may not be sufficient to support the claim.

There are some missing references for testing LLM's factuality.

SelfCheckGPT: Zero-Resource Black-Box Hallucination Detection for Generative Large Language Models
FActScore: Fine-grained Atomic Evaluation of Factual Precision in Long Form Text Generation
FacTool: Factuality Detection in Generative AI -- A Tool Augmented Framework for Multi-Task and Multi-Domain Scenarios

**Questions:**

See weaknesses

---

> ### Author Response · Authors · 2023-11-19
> **Author Response to Reviewer SmAT (Part 1/3)**
>
> We deeply appreciate the recognition given by the reviewers for the innovation and thoroughness of our work. We are also grateful for the constructive comments provided. We have responded to each suggestion regarding the weaknesses of the manuscript and will incorporate corrections in the subsequent version of the manuscript.
>
> > (1) Some points in the main results could be verified through better experiment settings, such as the point that "LLMs without instruction tuning underperform those with instruction tuning by 16.0%". It's better to conduct a peer-to-peer comparison since the instruction tuning is not the only factor of the two groups. For example, we can compare LLaMA with Alpaca and Vicuna, OPT vs. OPT-IML, T5 vs. Flan-T5, BLOOM vs. BLOOMz. I am not saying that we need to compare all of them, but the comparison could be made between the instruction tuning model and its backbone
>
>
> Thank you for your constructive feedback. As you recommended, we have conducted additional analyses to compare models with their instruction-tuned counterparts in Appendix A.9.
>
> The comparisons between LLaMA and its instruction-tuned versions, Alpaca and Vicuna, can be found in Table 2. Furthermore, we have conducted extra tests under the few-shots with CoT setting for T5-11B vs. Flan-T5-11B and BLOOM-6.7B vs. BLOOMz-6.7B. For T5, the accuracy was 18.6%, and the Macro F1 was 25.2%. In contrast, as shown in Table 2, Flan-T5 achieved an accuracy of 38.4% and a Macro F1 of 38.4%. Similarly, BLOOM’s performance was at an accuracy of 6.6% and Macro F1 of 12.2%, whereas BLOOMz showed a marked improvement with an accuracy of 27.5% and a Macro F1 of 27.7%.
> |             | Multifaceted |      | Structural |      | Adversarial |      | Temporal |      | Real-World |      | Domain-Specific |      | Multi-lingual |      | Overall |      |
> | :---------: | ------------ | ---- | ---------- | ---- | ----------- | ---- | -------- | ---- | ---------- | ---- | --------------- | ---- | ------------- | ---- | ------- | ---- |
> |             | ACC.         | F1   | ACC.       | F1   | ACC.        | F1   | ACC.     | F1   | ACC.       | F1   | ACC.            | F1   | ACC.          | F1   | ACC.    | F1   |
> |  LLaMA-7B   | 38.3         | 33.9 | 44.1       | 32.1 | 43.2        | 46.1 | 41.6     | 30.0 | 26.4       | 26.3 | 23.6            | 25.0 | 27.8          | 27.7 | 35.3    | 31.4 |
> |  Alpaca-7B  | 38.6         | 28.8 | 48.0       | 23.6 | 46.4        | 35.1 | 49.6     | 26.1 | 24.5       | 19.9 | 42.9            | 26.8 | 24.2          | 17.7 | 39.4    | 26.2 |
> |  Vicuna-7B  | 44.2         | 36.0 | 49.7       | 36.3 | 59.0        | 59.2 | 50.1     | 37.6 | 49.0       | 41.8 | 44.3            | 38.6 | 46.7          | 43.1 | 48.5    | 40.6 |
> |   T5-11B    | 26.3         | 31.3 | 29.2       | 27.6 | 16.5        | 25.1 | 24.9     | 28.5 | 2.2        | 2.9  | 3.9             | 7.3  | 27.7          | 18.1 | 18.6    | 25.2 |
> | Flan-T5-11B | 49.2         | 49.4 | 43.5       | 33.7 | 54.7        | 56.6 | 31.6     | 30.6 | 31.1       | 29.4 | 35.6            | 34.6 | 23.3          | 14.4 | 38.4    | 38.4 |
> | BLOOM-6.7B  | 10.7         | 13.5 | 0.8        | 3.5  | 2.0         | 3.7  | 3.7      | 7.7  | 5.4        | 8.5  | 11.8            | 15.6 | 9.8           | 15.9 | 6.6     | 12.2 |
> | BLOOMz-6.7B | 34.2         | 27.3 | 32.6       | 27.4 | 38.2        | 42.4 | 27.2     | 21.4 | 17.8       | 24.0 | 24.8            | 24.0 | 17.4          | 21.6 | 27.5    | 27.7 |
>
> These peer-to-peer comparisons consistently demonstrate that models undergoing instruction tuning generally outperform their non-tuned backbones. With few exceptions (e.g., LLaMA vs. Alpaca in terms of Macro F1), we observed an average performance improvement of 11.3% across all compared models. This indicates that instruction tuning has a significant positive impact on the performance of LLMs.

---

> ### Author Response · Authors · 2023-11-19
> **Author Response to Reviewer SmAT (Part 2/3)**
>
> > (2) For the "Using the CoT method, we observed a relative boost in performance in LLMs subjected to instruction tuning and RLHF", the tested models may not have strong CoT ability. It's good to see LLaMA-2 or GPT-4 and other recent models that claim to have strong reasoning abilities. Since CoT is not a generic ability that every LLM has.
>
> Thank you for your valuable comments. To address your query, we specifically tested the performance of GPT-4-0613 on the Pinocchio dataset under the "Few shots with CoT" setting. **In this context, GPT-4 achieved an accuracy of 49.9% and a Macro F1 score of 47.2%. These scores represent the best performance across all tested models, showing an average improvement of 2.1% over GPT-3.5-Turbo.**
>
> |                    |Multifaceted |         | Structural |          | Adversarial |          | Temporal |          | Real-World |          | Domain-Specific |          | Multi-lingual |          |
> | :--------------: | ----------- | -------- | ---------- | -------- | ----------- | -------- | -------- | -------- | ---------- | -------- | --------------- | -------- | ------------- | -------- |
> |                  | ACC.        | F1       | ACC.       | F1       | ACC.        | F1       | ACC.     | F1       | ACC.       | F1       | ACC.            | F1       | ACC.          | F1       |
> | Text-Davinci-002 | 47.7        | 47.7     | **50.8**   | 38.4     | 64.2        | 64.3     | 33.9     | 31.1     | **51.7**   | 41.4     | 36.4            | 36.1     | 43.1          | 39.5     |
> | Text-Davinci-003 | 51.1        | 47.8     | 44.3       | 33.7     | 64.1        | 63.7     | 41.4     | 35.1     | 48.0       | 42.8     | 40.4            | 41.4     | **43.7**      | **43.6** |
> | GPT-3.5-Turbo    | 53.6        | **53.1** | 44.8       | 37.8     | 67.4        | 67.4     | 37.4     | 33.9     | 50.4       | 43.1     | 38.7            | 40.3     | 41.3          | 41.1     |
> | GPT-4            | **56.4**    | 51.3     | 43.2       | **38.6** | **73.7**    | **73.3** | **45.2** | **39.6** | 49.9       | **45.3** | **50.4**        | **49.1** | 41.8          | 41.3     |

---

> ### Author Response · Authors · 2023-11-19
> **Author Response to Reviewer SmAT (Part 3/3)**
>
> >(3) Besides, you can include an additional metric to report the cases that LLMs failed to generate meaningful answers. Since LLM may refuse to answer according to security or other customized policies, treating all unanswered questions as wrong predictions may be unfair.
>
> During our experiments, we observed that only a negligible fraction of questions were left unanswered by the LLM due to security or other customized policies. Specifically, when using GPT-3.5-turbo as an example, out of 20,713 queries, fewer than 20 (< 1‰) responses were cases of refusal to answer. The main reason is that we have already excluded claims that could cause safety issues during the annotation progress.
>
> >(4) Sec-4.2 could be improved, especially the discussion around temporal analysis, " GPT-3.5-Turbo exhibits superior performance when dealing with outdated data as compared to updated data.", I think the numbers are close, and it may not be sufficient to support the claim.
>
> We have revised this section to reflect a more cautious and accurate statement that aligns better with the observed data. We appreciate your guidance in improving the clarity and precision of our discussion.
>
> >(5) There are some missing references for testing LLM's factuality.
>
> Thank you for your constructive suggestion. We have included all these references in the related work section (Page 9).

---

> > ### Author Response · Authors · 2023-11-23
> > **Summary of response and look forward to the feedback**
> >
> > We sincerely appreciate your constructive feedback on our manuscript. Based on your suggestions, we have made several enhancements and clarifications, which we believe have significantly improved the quality of our work.
> >
> > - Enhanced Model Comparison: We've enriched the manuscript with detailed comparisons between instruction-tuned models and their original versions. This new analysis in Appendix A.9 clearly illustrates the performance boost achieved through instruction tuning.
> >
> > - CoT Method Analysis: Responding to your insights, we've conducted additional tests on GPT-4 using the CoT method. The results underscore GPT-4's advanced reasoning capabilities, setting a new benchmark for performance in our tests.
> >
> > - Addressing Unanswered Questions: We've delved into the occurrence of unanswered queries due to security or policy restrictions. Our findings show that such instances are extremely rare, thus having a negligible effect on the overall performance metrics.
> >
> > - Refined Temporal Analysis in Sec-4.2: We took your advice to heart and revised Section 4.2, ensuring our discussion on temporal analysis is more aligned with the data and less prone to overinterpretation.
> >
> > - Inclusion of Critical References: Following your suggestion, we've added key references to the related work section. These additions provide a richer context and deeper understanding of the current landscape in LLM factuality research.
> >
> > We value your feedback and have strived to address each point thoughtfully. We're eager to hear your thoughts on these revisions.

---

### Official Review · Reviewer_9WHz · 2023-10-29

**Soundness:** 3 good
**Presentation:** 3 good
**Contribution:** 3 good
**Rating:** 6
**Confidence:** 3

**Summary:**

This paper proposes a benchmark named Pinocchio to research if LLMs learn factual knowledge correctly from seven aspects. The authors gather the text containing rich factual knowledge from existing datasets. Questions and answers are annotated manually based on the gathered text. With this proposed benchmark, the authors evaluate current popular LLMs.

**Strengths:**

Identifying the drawbacks of LLMs in terms of the learned factual knowledge is important. The proposed benchmark contains questions from different aspects, which could encourage researchers to find the specific issues of LLMs and explore how to address them.

**Weaknesses:**

1. Data leaky may happen. Note that the questions and answers are annotated based on the text of previous public datasets. The pretraining data of evaluated LLMs is likely to contain the text. This may cause data leaky, and LLMs might have memorized/learned shortcuts to answer the question. It makes the evaluation results on the benchmark less convincing.

2. Overclaimed contribution (as well as title) to the research question. I think the main contribution of this paper is the newly proposed benchmark. In the experiment, the authors research the extent and scope of factual knowledge within LLMs. However, the authors only test LLMs on the benchmark. There are no baselines (LLMs, datasets) as a comparison. Only showing accuracy numbers is not sufficient to give the answer to this research question.

**Questions:**

1. May I ask how you design the seven aspects for evaluation? Any high-level intuition?

2. Can 92.4% annotation accuracy (as well as 85.6% inter-annotator agreement rate) promise the quality of the dataset? Does that mean when the test accuracy of LLMs is larger than ~= 85%, it's meaningless to use this benchmark for evaluation?

---

> ### Author Response · Authors · 2023-11-19
> **Author Response to Reviewer 9WHz (Part 1/2)**
>
> We deeply appreciate the recognition given by the reviewers for the innovation and thoroughness of our work. We are also grateful for the constructive comments provided. We have responded to each suggestion regarding the weaknesses of the manuscript and will incorporate corrections in the subsequent version of the manuscript.
>
> > (1) Data leaky may happen. Note that the questions and answers are annotated based on the text of previous public datasets. The pretraining data of evaluated LLMs is likely to contain the text. This may cause data leaky, and LLMs might have memorized/learned shortcuts to answer the question. It makes the evaluation results on the benchmark less convincing
>
> Thank you for raising the critical issue of data leakage. We have conducted several measures to alleviate data leakage.
> - We transformed the original dataset's statements into a question-and-answer (QA) format to mitigate this concern. This adaptation reduced the likelihood of data leakage.
> - It is also worth noting that the labels from the original fact-checking dataset, typically categorized as "Refutes" or "Supports," do not directly apply to our new dataset. In the original context, these labels indicate whether the provided evidence supports or contradicts a given claim. The model determines whether a factual question is "Factual" or "Non-Factual" based on its inherent knowledge.
> - Furthermore, we manually revised their labels for questions that underwent factual changes, and around 7% of the statements experienced such factual alterations.
>
> Lastly, we conducted experiments to evaluate the data contamination. Following the methodology proposed in [1], we evaluated the claim in the original dataset and found a data leakage rate of 21.3%. After reannotating the original claims into question-answer pairs manually (by annotators), this rate decreased to 7.4%. We compare this rate with other datasets below. When compared with other benchmarks, our benchmark has a significantly lower contamination rate.
>
> | Dataset | Contamination Rate |
> | :-----| ----: |
> | MMLU | 29.8% |
> | HellaSwag | 25.6% |
> | ARC | 16.3% |
> | Ours | **7.4%** |
>
> *Reference:*
>
> [1] An Open Source Data Contamination Report for Llama Series Models. ArXiv 2023.
>
> > (2) Overclaimed contribution (as well as title) to the research question. I think the main contribution of this paper is the newly proposed benchmark. In the experiment, the authors research the extent and scope of factual knowledge within LLMs. However, the authors only test LLMs on the benchmark. There are no baselines (LLMs, datasets) as a comparison. Only showing accuracy numbers is not sufficient to give the answer to this research question
>
>
> To address your concern,  we have now provided performance results for GPT-3.5-Turbo in the few shots with CoT settings on various datasets, as shown in the table below.
> Due to time and resource constraints, we will gradually include more LLMs for comparison in future revisions of the paper.
>
> | Datasets    | Accuracy | F1   |
> | ----------- | -------- | ---- |
> | FEVER       | 56.3     | 55.7 |
> | Feverous    | 36.4     | 31.0 |
> | FoolMeTwice | 60.2     | 61.1 |
> | VitaminC    | 42.8     | 42.7 |
> | Politifact  | 36.2     | 35.5 |
> | PubHealth   | 56.3     | 49.5 |
> | X-Fact      | 35.8     | 33.1 |
>
> > (3) May I ask how you design the seven aspects for evaluation?  Any high-level intuition?
>
> Thank you for your question regarding the design of the seven evaluation aspects. **To develop these aspects, we first undertook a comprehensive survey of numerous existing fact-checking datasets, the summary of which is presented in Appendix A.6, Table 9. ** Our analysis focused on identifying the various challenges these datasets aim to address regarding factuality.
>
> We found that the challenges tackled by these datasets generally align with the seven aspects we have listed in our dataset. These aspects appear individually or in combination across the surveyed datasets, indicating their relevance and importance in fact-checking. This realization led us to intentionally design our evaluation framework around these seven specific challenges, ensuring that our benchmark is not only comprehensive but also directly addresses the core difficulties encountered in current fact-checking tasks.

---

> ### Author Response · Authors · 2023-11-19
> **Author Response to Reviewer 9WHz (Part 2/2)**
>
> > (4) Can 92.4% annotation accuracy (as well as 85.6% inter-annotator agreement rate) promise the quality of the dataset? Does that mean when the test accuracy of LLMs is larger than ~= 85%, it's meaningless to use this benchmark for evaluation?
>
> The 92.4% annotation accuracy and 85.6% inter-annotator agreement rate indeed signify a high-quality dataset. Notably, the primary source of disagreement among annotators was the "Not Enough Information" (NEI) label. According to research by Guo et al. [1], defining the NEI category is inherently challenging, which can account for some of the disagreements.
> Moreover, compared to commonly accepted benchmarks in the field, our dataset's quality is notably superior. For instance, the FEVER [2] dataset has an inter-annotator agreement rate of 0.68, LIAR [3] is at 0.82, and FEVEROUS [4] stands at 0.65. Our dataset surpasses these widely used benchmarks, underscoring its reliability.
>
> *References:*
>
> [1] A Survey on Automated Fact-Checking. TACL 2021
>
> [2] FEVER: a Large-scale Dataset for Fact Extraction and VERification. NAACL 2018
>
> [3] “Liar, Liar Pants on Fire”: A New Benchmark Dataset for Fake News Detection. ACL 2017
>
> [4] FEVEROUS: Fact Extraction and VERification Over Unstructured and Structured information. NeurIPS 2021

---

> > ### Author Response · Authors · 2023-11-23
> > **Refinements in Response to the Issue of Overclaiming**
> >
> > Thank you for your constructive feedback which has guided us in enhancing our manuscript.
> >
> > We understand the criticality of presenting our study's scope and limitations with utmost precision. In light of this, we have further refined the verbiage throughout our manuscript, encompassing the title, abstract, introduction, methodology, experiments, related work, and conclusions. This additional layer of revision is intended to enhance the accuracy in portraying our research contributions and results. To ensure these modifications are easily identifiable, we have distinctly marked them in red in the updated version of our paper.

---

### Official Review · Reviewer_1LZe · 2023-11-04

**Soundness:** 3 good
**Presentation:** 3 good
**Contribution:** 3 good
**Rating:** 5
**Confidence:** 4

**Summary:**

1. Lack of datasets to evaluate LLMs ability to hold and reason over factual knowledge is the primary motivation of the paper to come up with a new benchmark called Pinocchio benchmark. This benchmarks can evaluate LLMs in tasks that needs to know factual knowledge from diverse set of domains, multiple facts to reason, and the facts may be present in different information sources. The work also performs varied experiments and analysis on factuality or non factuality for multiple large language models. The experiments and conclusions are presented well.
    2. Datasets:
        1. Most datasets have evidence but Pinnachio only cares yes or no answers without any evidence, specifically multiple choice question answers. More explanation on how this is a good strategy to conclude if LLMs are able to memorize and use facts is unclear from the paper.
        2. There is no clear motivation in the paper on why these datasets were chosen — are these the only specific 7 categories of questions? Or was it the distribution of the known datasets and the data points?
        3. Evaluating answers with evidences has led to better understanding on the performance of large language models. It will be good to compare performances of the large language models on the datasets vs how they are transformed for this setting
    3. Conclusions from this work:
        1. The work also performs varied experiments and analysis on factuality or non factuality and comes up with the following conclusions:
            1. Instruction tuning performs better than non-instruct tuning
            2. Few shot with COT > Few shot > Zero-shot with COT > Zero Shot
            3. The number of hops in multi-hop reasoning dictates the performance decrease of GPT
            4. LLMs have limited ability to use structured data
            5. Numerical reasoning is harder for LLMs
            6. LLMs are unable to use the latest data (temporal) but are able to answer with older data (trained on)
        2. Comments:
            1. These conclusions hold for Large Language Models in a more generic sense rather than just factuality. It’s important for the authors to clarify the important of new conclusions in comparison on what already exists. Explanations of how these are different when made on the same datasets before deriving them to a multiple-choice QA dataset.

**Strengths:**

1. An annotated dataset of 20k that can evaluate factuality of large language models.
2. Work is presented well with experiments and conclusions.

**Weaknesses:**

1. Data is derived from existing datasets but the motivation for transformation is not clearly specified
2. Most conclusions of the experiments are already made in isolation prior to this work. The novelty in experiments and conclusion should be well specified.

**Questions:**

Already in the summary

---

> ### Author Response · Authors · 2023-11-19
> **Author Response to Reviewer 1LZe (Part 1/3)**
>
> We deeply appreciate the recognition given by the reviewers for the innovation and thoroughness of our work. We are also grateful for the constructive comments provided. We have responded to each suggestion regarding the weaknesses of the manuscript and will incorporate corrections in the subsequent version of the manuscript.
>
> >(1) Most datasets have evidence but Pinnachio only cares yes or no answers without any evidence, specifically multiple choice question answers. More explanation on how this is a good strategy to conclude if LLMs are able to memorize and use facts is unclear from the paper.
>
> Thanks for the insightful question.
>
> We wish to clarify that there is a distinct separation in motivations between the existing fact-checking datasets and our Pinocchio benchmark. **The primary objective of traditional fact-checking datasets is to support the development of automated fact-checking systems**. To verify a claim, journalists typically must search through numerous sources to uncover pertinent evidence, assess the credibility of those sources, and synthesize a verdict based on the gathered evidence. To mimic this complex process, such datasets are designed to function within a framework comprising three essential components: claim detection, evidence retrieval, and claim verification, with evidence being a pivotal factor in these systems.
>
> **In contrast, our Pinocchio benchmark is designed primarily to assess the breadth and depth of factual knowledge contained within LLMs.** Unlike the datasets mentioned above, Pinocchio is not intended to assist in creating automated fact-checking systems that emulate the procedures carried out by professional fact-checkers. Additionally, we focus on evaluating the factual knowledge 'inside' LLMs. We are less concerned with an LLM's ability to process and infer from 'external evidence' to formulate conclusions.
>
> Pinocchio aligns more closely with initiatives that utilize language models as a knowledge base [3] by employing multiple-choice formats or that aim to refine the information stored within a language model by implementing boolean answers [4]. Previous research [5] indicated that LLMs demonstrate robust calibration when presented with multiple-choice questions, drawing out the model's inherent knowledge. Our experiment supports this finding; as demonstrated in Table 5, posing questions as inputs yields a superior extraction of factual knowledge compared to the declarative claims in the original datasets, resulting in a 2.3% increase in performance.
>
> *References:*
>
> [1] Automated Fact-Checking for Assisting Human Fact-Checkers. IJCAI 2021
>
> [2] A Survey on Automated Fact-Checking. TACL 2021
>
> [3] Language Models as Fact Checkers? ACL 2020
>
> [4] Editing Factual Knowledge in Language Models. EMNLP 2021
>
> [5] Language Models (Mostly) Know What They Know. Anthropic Technical Report 2022
>
> >(2) There is no clear motivation in the paper on why these datasets were chosen — are these the only specific 7 categories of questions? Or was it the distribution of the known datasets and the data points?
>
> **As detailed in Appendix A.6, Table 9, our selection was based on an extensive survey of existing fact-checking datasets.** This survey aimed to identify and summarize the different challenges these datasets address associated with factuality.
>
> We discovered that these datasets commonly target the seven question categories outlined in our study. These categories, appearing individually or in various combinations in the datasets we reviewed, represent the most significant and prevalent challenges in current fact-checking tasks. Based on this comprehensive analysis, we deliberately designed our benchmark to encompass these seven specific challenges, ensuring that our evaluation framework is both relevant and comprehensive.

---

> ### Author Response · Authors · 2023-11-19
> **Author Response to Reviewer 1LZe (Part 2/3)**
>
> >(3) Evaluating answers with evidences has led to better understanding on the performance of large language models. It will be good to compare performances of the large language models on the datasets vs how they are transformed for this setting.
>
> Thank you for your suggestion regarding the evaluation of LLMs using evidence-based answers. In the few shots with the CoT setting, LLMs provide outputs that include their reasoning processes. We manually sampled and analyzed 100 instances of these reasoning processes, comparing them with the gold evidence.
>
> Of these 100 instances, 38 showed correct reasoning processes, leading to accurate conclusions by the LLMs. However, 51 instances had incorrect reasoning processes, resulting in erroneous conclusions. Interestingly, in 4 instances, the reasoning process was flawed, but the LLMs still arrived at the correct answer. Conversely, there were 6 instances where the reasoning was accurate, but the LLMs reached incorrect conclusions. Additionally, we encountered a unique case where the LLM's reasoning process was internally contradictory.
>
> **A more detailed case study of these observations is included in Appendix A.7 (pages 30-31).**
>
> >(4) It’s important for the authors to clarify the important of new conclusions in comparison on what already exists.
>
> Thank you for your thoughtful feedback. Please allow us to clarify and highlight the important findings we had based on the proposed benchmark:
>
> - **Divergent Responses to Prompting Techniques:** We've observed that the efficacy of prompting techniques varies significantly between complex reasoning and factual knowledge tasks. Advanced prompting strategies like increased shot counts, complex Chains of Thought (CoT), and self-consistency show promise in reasoning tasks including mathematics reasoning, logical reasoning, symbolic reasoning, etc. However, these methods fall short in factual queries (refer to Table 5). Our conjecture is that this divergence might stem from fundamental differences in how LLMs handle factual-based tasks versus those requiring complex reasoning. In factual-based tasks, the LLMs should first remember the facts correctly and then perform reasoning on the relevant facts. If the facts are wrong in the first place, simply enhancing the reasoning is not helpful. Prior prompting techniques for reasoning can not help LLMs to better elicit factual knowledge.
>
> - **LLMs' Overconfidence in Facts:** LLMs display a higher level of confidence in their answers to factual questions, which can be misleading. Despite multiple iterations of the same question (as self-consistency prompting suggests), LLMs often replicate their errors (Table 5). After human inspections, we found that LLMs tend to make the same factual mistakes no matter whether we ask them how many times or adopt a decoding strategy to increase the diversity of answers.
>
> - **Self-Reflection in Correcting Factual Errors:** Self-refinement prompts LLMs to reassess their answers, showing significant improvements for factual questions (Table 5). After manual inspection, we found that when LLMs re-evaluate their earlier responses with CoTs, they can sometimes detect and rectify errors within their own reasoning pathways. This finding is illustrates a potential method for tempering overconfidence and enhancing fact-based accuracy in LLM responses.
>
> - **The Interplay of Structured and Unstructured Knowledge:** Contrary to initial assumptions that mixing structured (tables) and unstructured (text) knowledge sources complicates answering factual questions, our study reveals that LLMs are adept at integrating the two. This blend can even facilitate the recall of structured knowledge (Figure 3(b)). This equivalency between structured and combined knowledge sources is significant because it suggests that LLMs may use textual contexts to reinforce and access tabular facts, a valuable consideration for knowledge representation and retrieval strategies.
>
> - **Challenges in Factual Precision:** LLMs exhibit proficiency in distinguishing between entities but struggle with precise numerical information (Figure 3(c)). Our manual analyses suggest that although LLMs effectively memorize and differentiate entity details, they have difficulties with exact figures like numbers and dates. This distinction is crucial for understanding LLM limitations in tasks requiring precision and lays the groundwork for focused improvements in numeral cognition for LLMs.

---

> ### Author Response · Authors · 2023-11-19
> **Author Response to Reviewer 1LZe (Part 3/3)**
>
> >(5) Explanations of how these are different when made on the same datasets before deriving them to a multiple-choice QA dataset.
>
> Thank you for the insightful comment. Please let us elaborate on the nuances that distinguish the conversion to a QA format:
> - **Dataset Update and Factuality Assessment:** Facts and data can evolve over time. Therefore, as part of our preparation, we reassess and update the original dataset to ensure the accuracy and timeliness of the information. During this step, approximately 7% of statements were found to have changed in their degree of truthfulness or accuracy.
> - **Addressing Safety and Bias Concerns:** (LLMs) are sometimes preprogrammed to avoid engaging with content that violates certain safety or ethical guidelines. To circumvent any potential issues related to these models rejecting content during the analysis, we thoroughly inspect all items in the dataset. This review process includes discarding any items (4%) that may provoke controversy or be deemed unsafe, thereby streamlining the dataset to focus on informative and neutral content that LLMs can handle without contravening their operating principles.
> - **Eliciting Factual Knowledge:** As reflected in prior studies, presenting questions in a multiple-choice format can effectively harness the knowledge calibration of LLMs [1]. We've observed, and Table 5 on page 8 highlights, that this approach enhances the LLMs' capability to draw upon its internal factual knowledge—delivering a measurable improvement of 2.3% in accuracy. Moreover, we manually sampled and analyzed 100 instances of reasoning processes under the QA format, revealing that presenting information in a QA format, more aligned with human reasoning patterns, significantly enhances the inference capabilities of LLMs. A more detailed case study of these observations is included in Appendix A.7 (pages 30-31).
>
> References:
>
> [1] Language Models (Mostly) Know What They Know. Anthropic Technical Report  2022

---

> > ### Author Response · Authors · 2023-11-23
> > **Summary of response and look forward to the feedback**
> >
> > We greatly appreciate the thoughtful critique and suggestions from Reviewer 1LZe. Below is a summary of our revisions and clarifications based on the provided feedback.
> > - Clarification on Pinocchio Benchmark's Approach: We've addressed the concern about Pinocchio benchmark focusing on yes/no answers without evidence. Our benchmark aims to evaluate the intrinsic factual knowledge within LLMs, distinct from traditional fact-checking datasets. We argue that this approach better assesses the factual knowledge embedded in LLMs, as supported by research showing LLMs' robust calibration in multiple-choice settings.
> > - Rationale Behind Dataset Selection: We've elaborated on why specific datasets were chosen, emphasizing our comprehensive analysis of existing fact-checking datasets. This led us to focus on seven major categories of questions, representing prevalent challenges in current fact-checking tasks.
> > - Comparing Performances with Evidence-based Answers: Responding to the suggestion of comparing LLM performances with and without evidence-based answers, we analyzed instances where LLMs provided reasoning processes. This comparison highlights interesting findings about LLMs' reasoning accuracy and the effectiveness of their conclusions.
> > - New Conclusions and Their Significance: We clarified the importance of our new conclusions, detailing how our findings differ from existing research. These include observations on prompting techniques, LLMs' overconfidence in facts, the potential for self-reflection in correcting factual errors, and challenges in factual precision.
> > - Transforming to a Multiple-Choice QA Dataset: We provided more details on the nuances involved in converting the original datasets to a QA format. This includes updates for factual accuracy, addressing safety and bias concerns, and enhancing the elicitation of factual knowledge from LLMs.
> >
> > We hope these revisions and clarifications address your concerns and look forward to any additional feedback or questions.

---

### Official Review · Reviewer_XTL4 · 2023-11-06

**Soundness:** 3 good
**Presentation:** 3 good
**Contribution:** 3 good
**Rating:** 8
**Confidence:** 4

**Summary:**

This paper introduces a benchmark that specifically evaluates the parametric knowledge acquired by LLMs during pretraining. The authors collected and recast existing knowledge-intensive datasets such as fact checking (e.g., FEVER) across seven domains:
- Multiple-facts (e.g., multiple supporting facts)
- Structural (e.g., with tables)
- Adversarial (e.g., unrelated facts)
- Temporal (e.g., Wikipedia revisions)
- Real-world (e.g., politifact)
- Domain-specific (e.g., medical and science)
- Multi-lingual (e.g., French, Chinese, and more)
All examples are formulated as multiple-choice problems with three classes. 10 human annotators rewrote the original claims into questions while preserving the original facts and their labels.

The experiment setup primarily focuses on LLMs with prompting (zero-shot, few-shot, w/CoT). The major pretrained LLMs are evaluated on this benchmark, including public LLMs such as OPT, BLOOM, and LLaMA and commercial LLMs such as GPT3 and ChatGPT. Overall, ChatGPT and GPT3 outperform other LLMs by large margin, regardless of the prompting methods used. The fine-grained results show that the commercial LLMs are strong on many domains but underperform other LLMs on the temporal domain. Additionally, the authors performed analysis on different aspects such as question types and performance.

**Strengths:**

- This paper investigates factual knowledge in LLMs, which is a well-motivated problem.
- A comprehensive benchmark covering different aspects of factual knowledge is proposed, particularly suitable for prompting.

**Weaknesses:**

Dataset Design: _“...multi-choice questions are a simple but good proxy to evaluate the potential of advanced abilities of LLMs…”_: I think this claim should be supported by prior work. I tend to disagree with this statement as Chen and Durrett (2019) concludes that, for some datasets, multiple-choice questions could be easily gamed (i.e., spurious correlations between questions and labels).  Link: https://arxiv.org/pdf/1904.12106.pdf

A significant number of related papers have not been cited. There are numerous strands of prior work that investigate the factual knowledge and reasoning abilities of large language models (LLMs), including multi-hop reasoning, temporal and geographical questions, new entities, and complex inference. (I’m listing a few of them here)
- Did Aristotle Use a Laptop? A Question Answering Benchmark with Implicit Reasoning Strategies (https://arxiv.org/abs/2101.02235)
- SituatedQA: Incorporating Extra-Linguistic Contexts into QA (https://arxiv.org/abs/2109.06157)
- A Dataset for Answering Time-Sensitive Questions (https://arxiv.org/abs/2108.06314)
- Entity Cloze By Date: What LMs Know About Unseen Entities (https://arxiv.org/abs/2205.02832)
- RealTime QA: What's the Answer Right Now? (https://arxiv.org/abs/2207.13332)
- StreamingQA: A Benchmark for Adaptation to New Knowledge over Time in Question Answering Models (https://arxiv.org/abs/2205.11388)
- Can LMs Learn New Entities from Descriptions? Challenges in Propagating Injected Knowledge (https://arxiv.org/abs/2305.01651)
- FreshLLMs: Refreshing Large Language Models with Search Engine Augmentation (https://arxiv.org/abs/2310.03214)

**Questions:**

- _“...but in Pinocchio, we only need to judge the factuality of the question.”_ Does this mean all supporting facts are not included in the questions?

Minor:
- The terms “Multifaceted” and “multifacted” are used interchangeably. Is this a typo?

---

> ### Author Response · Authors · 2023-11-19
> **Author Response to Reviewer XTL4 (Part 1/2)**
>
> Thanks for the insightful questions.
>
> >(1)The claim "...multi-choice questions are a simple but good proxy to evaluate the potential of advanced abilities of LLMs…" should be supported by prior work.
>
> Multiple-choice questions offer a practical approach to assessing current LLMs' complex capabilities, of which GPT-4 is a prime example [1]. Key benchmarks such as the MMLU [2], HellaSwag [3], ARC [4], BIG-Bench Hard [5], and TruthfulQA [6], all of which utilize multi-choice formats, serve distinct purposes in evaluating various aspects of GPT -4's proficiency. Precisely, the MMLU gauges an LLM's academic and professional knowledge breadth and depth. HellaSwag is designed to test commonsense reasoning, and ARC focuses on one's ability to tackle challenging questions. BIG-Bench Hard presents complex reasoning tasks that challenge even the most advanced LLMs, while TruthfulQA measures how LLMs mimic human falsehoods.
> Furthermore, the evaluation of language generation brings its own challenges, as a universal metric for measurement is currently lacking [7], which multiple-choice questions help mitigate by offering straightforward classification accuracy for assessment [1]. Also, prior studies [8] underscore that LLMs demonstrate reliable calibration when faced with multiple-choice scenarios, suggesting that this format is adept at extracting a model's latent knowledge.
> In response to your insight, we have included this discussion in our paper, and rewritten section 2.2 **ANNOTATION AND QUALITY CONTROL** in the rebuttal version.
>
> *References:*
>
> [1] GPT4 Research Report. https://openai.com/research/gpt-4
>
> [2] Measuring Massive Multitask Language Understanding. ICLR 2021
>
> [3] HellaSwag: Can a Machine Really Finish Your Sentence? ACL 2019
>
> [4] Think you have Solved Question Answering? Try ARC, the AI2 Reasoning Challenge. ArXiv 2018
>
> [5] Challenging BIG-Bench Tasks and Whether Chain-of-Thought Can Solve Them. ACL 2023
>
> [6] TruthfulQA: Measuring How Models Mimic Human Falsehoods. ACL 2022
>
> [7] A Survey of Evaluation Metrics Used for NLG Systems. ACM CSUR 2022
>
> [8] Language Models (Mostly) Know What They Know. Anthropic Technical Report 2022
>
> >(2) According to Chen and Durrett (2019), multiple-choice questions could be easily gamed (i.e., spurious correlations between questions and labels).
>
> Chen and Durrett (2019) focused on multi-hop question-answering datasets, where a model is required to reason over multiple supporting documents to answer the question. In order to verify that it is possible to pick the correct answer without consulting the related documents, they construct a "no context" baseline. This "no context" baseline achieves higher performance on the multiple-choice WikiHop dataset, comparable to some state-of-the-art QA systems. This result shows that WikiHop can solve reasonably well without using multiple documents, indicating that the dataset is potentially gamed.
>
> Following Chen and Durrett (2019), we developed the same "no context" baseline to investigate the spurious correlations between questions and labels in our dataset. The results are shown below:
>
> | Method | Performance |
> | :-----| ----: |
> | No Context | **28.3** |
> | LLaMA-7B | 31.6 |
> | Alpaca-7B | 37.8 |
> | Vicuna-13B | 45.2 |
> | GPT-3.5 | 47.0 |
>
> Our experimental findings show that our dataset does not exhibit the same level of vulnerability to the exploitation of question-label correlations as observed by Chen and Durrett (2019) in the WikiHop dataset. With performance improvements of 16.9 points by Vicuna-13B and 18.7 by GPT-3.5 over the "no context" baseline, our results offer compelling evidence that our dataset is more resilient to such biases, contrary to the reported susceptibilities within WikiHop.
>
> We extended our analysis to include a direct comparison with several established multiple-choice question-answering benchmarks, such as the WikiHop mentioned above, as well as with other prevalent benchmarks like TruthfulQA and ARC utilized in evaluating LLMs. The performance of the "no context" baseline across these benchmarks is displayed in the table below:
>
> | Dataset | Performance |
> | :-----| ----: |
> | WikiHop | 59.7 |
> | TruthfulQA | 34.5 |
> | ARC | 33.2 |
> | Ours | **28.3** |
>
> Evidently, our proposed dataset presented the most challenge to the "no context" baseline, marking the lowest performance compared to other datasets. The notable performance on WikiHop, with a "no context" baseline score of 59.7%, underscores the presence of spurious correlations that facilitate gaming that dataset. On the contrary, the lower baseline performances on TruthfulQA and ARC suggest that such issues are less prevalent. Our dataset, therefore, not only stands out as the least prone "to be gamed" but also underscores its robustness and the high level of rigor needed to tackle it effectively.

---

> ### Author Response · Authors · 2023-11-19
> **Author Response to Reviewer XTL4 (Part 2/2)**
>
> >(3) A significant number of related papers have not been cited.
>
> Thank you for providing the list of the related work missing in the paper. We have included the paper you listed in the rebuttal version.
>
> Furthermore, we extensively survey other related efforts in these topics. **As shown in Table 13 in the Appendix (page 36), we comprehensively compared 37 research efforts. We also included a new section called RELATED WORK: QUESTION ANSWERING DATASETS in Appendix A.10 (pages 32-35), where we discuss a significant number of related work in detail (more than 50 references are added)**.
>
> >(4) “...but in Pinocchio, we only need to judge the factuality of the question.” Does this mean all supporting facts are not included in the questions?
>
> Apologies for any confusion earlier. Please allow us to clarify: in our proposed dataset, the focus is exclusively on assessing the factual accuracy of the questions themselves, without the inclusion of external supporting facts.
> In typical fact-checking datasets, labels like "Supports" or "Refutes" are used to signal whether external evidence confirms or denies a particular claim. These datasets are aimed at fostering the development of systems able to emulate the comprehensive process journalists go through to fact-check a claim, which involves sifting through various sources for relevant evidence and consequently reaching a conclusion regarding the claim's accuracy. This process relies heavily on evidence gathering, a critical aspect of traditional fact-checking operations.
>
> However, our Pinocchio benchmark is designed with the objective of gauging the factual knowledge stored within LLMs. The benchmark evaluates LLMs' capacity to reason using the factual knowledge they have previously assimilated, thus, intentionally excluding the provision of external supporting facts. In simple terms, we are testing the LLMs' ability to verify facts based solely on the information they have internalized without referencing additional evidence. We are less concerned with an LLM's ability to process and infer from 'external evidence' to formulate conclusions.
> To create this specific dataset, we have re-evaluated each statement from the original fact-checking datasets by employing annotators to research various sources and verify facts, which led us to re-annotate the claims. This method ensures that our dataset appropriately tests the LLMs' factuality without relying on the inclusion of supporting facts within the questions.
>
> > (5) Typo of “Multifaceted”.
>
> Thank you very much for pointing out this inconsistency. Indeed, "Multifacted" was a typographical error in our manuscript. We have revised the document and replaced all instances of "multifacted" with "multifaceted" to ensure consistency and accuracy in our terminology.

---

> > ### Comment · Reviewer_XTL4 · 2023-11-22
> >
> > I appreciate your clarification. While the authors have addressed several of my concerns, the paper's narrative remains somewhat misleading, as Reviewer 9WHz notes regarding overclaiming. This benchmark tests LLMs only on facts covered by existing datasets, which may include outdated information. I will adjust my score accordingly.

---

> > > ### Author Response · Authors · 2023-11-23
> > > **Thanks for your valuable feedback**
> > >
> > > Thank you for your insightful comments. We have taken them into serious consideration and would like to address your concerns as follows:
> > >
> > > > This benchmark tests LLMs only on facts covered by existing datasets, which may include outdated information.
> > >
> > > Thank you for your insightful comments regarding the coverage and timeliness of the facts in our dataset. Please allow us to clarify the outdated issue:
> > >
> > > - We have undertaken a comprehensive re-annotation process for the existing datasets to create our Pinocchio dataset. This re-annotation was based on the time points at which the re-annotation was conducted. As a result, we ensured that all labels in the Pinocchio dataset reflect the most current knowledge available at the time of this re-annotation.
> > > - Furthermore, it is important to note that, with the exception of the temporal domain factual questions, which have already been updated to the end of 2022, the majority of the questions in our dataset do not possess a time-sensitive nature. Thus, they do not necessitate updates to reflect the very latest information.
> > >
> > > > Issue of overclaiming
> > >
> > > We have meticulously revised the narratives used throughout our paper, including in the title, abstract, introduction, methodology, experiments, related work, and conclusion sections, to more precisely reflect our contributions and findings. These modifications are highlighted in red in the revised version of our paper to facilitate easy identification.

---

### Author Response · Authors · 2023-11-21
**Looking forward to discussing with reviewers**

Dear Reviewers,

Thank you very much for your invaluable feedback on our paper. We have meticulously reviewed each of your points and endeavored to address them thoroughly. We would greatly appreciate it if you could review our responses and let us know if you have any additional comments regarding the paper or the rebuttal. We are eager to embrace all your critiques and integrate them into our work.

Thank you!

---

### Meta-Review · Area_Chair_SbML · 2023-12-05

**Metareview:**

This paper introduces a new benchmark named Pinocchio, which helps evaluate the parametric knowledge acquired by LLMs during pretraining. The authors recast existing datasets (e.g., FEVER) to analyze LLM's fact-related capabilities across seven different facets and tasks (structural with tables, temporal with revisions, etc.). The experiments in the paper primarily focus on LLM prompting (e.g., zero-shot, CoT) and provide a comparison of various commercial (e.g., ChatGPT) and open-source LLMs (e.g., LLaMA) across these facets. The initial submission of the paper had some issues such as data leakage and overclaiming, but the authors provided extensive revisions and responses that appear to address all the main concerns (resulting in a data leakage rate of only 7.4%, significantly less than comparable datasets). The authors also conducted additional experiments on established multiple-choice question-answering benchmarks (e.g., WikiHop). Overall, the reviewers found the contribution of this paper to be quite significant, as a benchmark assessing various aspects of factual knowledge could be of great value to the community and research on knowledge-intensive tasks. In light of these contributions, I recommend accepting this paper. Note: The data leakage analysis provided in the rebuttal does not seem to appear in the revision of the paper, and I would recommend adding it to the camera-ready paper (e.g., in the appendix), as other readers might also wonder about possible data leakage.

**Justification For Why Not Higher Score:**

Considering the relatively large number related works and datasets available in this area (as seen in review XTL4), I would not raise my recommendation to a higher level of acceptance.

**Justification For Why Not Lower Score:**

The introduction of this new benchmark analyzing different facets factual knowledge in LLMs could be quite valuable to the community and research on knowledge-intensive tasks.

---

### Decision · Program_Chairs · 2024-01-16

Accept (spotlight)